

# Brief communication: Sensitivity analysis of peak water to ice thickness and temperature: A case study in the Western Kunlun Mountains of the Tibetan plateau

Lucille Gimenes[1], Romain Millan[1], Nicolas Champollion[1], and Jordi Bolibar[1]

[1]Institut des Géosciences de l'Environnement, Université Grenoble-Alpes, Grenoble, France

**Correspondence:** Lucille Gimenes (lucille.gimenes@univ-grenoble-alpes.fr) and Romain Millan
(romain.millan@univ-grenoble-alpes.fr)

**Abstract.** This study investigates the sensitivity of peak water in the western Kunlun Mountains of the Tibetan Plateau. Using the Open Global Glacier Model, we analyze how variations in inverted ice volume and temperature bias under different climate scenarios affect peak water timing and magnitude. We compare two global ice thickness datasets, revealing substantial differences in the predicted peak water timing and magnitude. The results highlight that smaller initial ice volumes lead to
earlier peak water occurrences, particularly under the SSP5-8.5 scenario. Temperature bias also significantly influences runoff magnitude and the timing of peak water, especially under high-emission scenarios. These findings underscore the importance of accurate ice thickness estimates and climate projections for predicting future water availability and informing water management strategies in glacier-dependent regions.

## 1 Introduction

Modeling the future evolution of glaciers is essential due to their significant impact on sea level rise ($0.61 \pm 0.08$ mm Sea Level Equivalent - SLE - $yr^{-1}$ for the 2006-2015 period) and freshwater resources (Nauels et al., 2017). Consequently, the projected changes in runoff will also impact downstream water management (Hock et al., 2019). With increasing air temperatures, glacier ablation and therefore glacier runoff is expected to rise and reach a maximum, defined as "peak-water," after which glacial freshwater outputs will decline due to the shrinking glacier area (Huss and Hock, 2018). Determining precisely the timing

and magnitude of maximum runoff is therefore of prime importance for freshwater management, likely affecting ecosystems, drinking water resources as well as other sectors as agriculture or hydro-power production (Arias et al., 2021).

The indirect inversion of ice thicknesses is a critical step which can have a substantial impact on the peak water (Huss and Farinotti, 2012). However, existing glacier thickness estimations are uncertain and exhibit notable differences in both the spatial distribution and the overall volume of glaciers (Farinotti et al., 2019; Millan et al., 2022). Advances in satellite remote sensing

have allowed the emergence of new global data products, such as the first global estimate of glacier mass change (Hugonnet et al., 2021) or global mapping of thicknesses based on surface flow velocities (Millan et al., 2022). Among available large-scale glacier models, flowline models like the Open Global Glacier Model (OGGM, Maussion et al., 2019) or the Global Glacier Evolution Model (GloGEM, Huss and Hock, 2015) are based on flowline versions of the Shallow Ice Approximation





(SIA, Hutter, 1983). On the other hand, models such as Elmer/Ice (Gagliardini et al., 2013) or the Ice Sheet System Model
(ISSM, Larour et al., 2012) work in three dimensions with a higher degree of complexity, especially concerning ice dynamics,
and numerous parameters to optimize (Zekollari et al., 2022). Simulations are therefore more computationally expensive,
and more challenging to apply on a large scale. The very recent development of emulators based on deep learning, or the
implementation of universal differential equations, holds strong potential to address these issues to better simulate glacier flow
dynamics (Jouvet, 2023; Bolibar et al., 2023) but not yet applicable at large scale. Therefore, despite the limitations of global
scale models (essentially flowline and approximation of ice dynamics), the growing developments and physics improvements,
along with their user-friendly nature, make them unique and useful tools for estimating the evolution of all worldwide glaciers
(Marzeion et al., 2018; Rounce et al., 2023; Zekollari et al., 2024).

In this study, we investigate the sensitivity of peak water timing, magnitude and duration, to different inverted ice volumes
and temperature bias.

## 2   Data and Methods

### 2.1   Region of interest

Our study focuses on the northern part of the Karakoram, precisely within the West Kunlun mountain range, situated at the
confluence of the Xinjiang Autonomous Region and the Tibetan Plateau (RGI Consortium, 2017). This study specifically
targets a group of 160 glaciers with a total surface area of approximately 2900 km$^2$ (Fig. 1). These are located at very high
elevations (5500-6400 m, Ke et al., 2015), in a region characterized by largely sub-zero temperatures, often reaching -10°C on
annual averages. This region has an icefield-like geometry that is hosting a large variety of glaciers. The northern part of the
selected region has a steeper terrain, with mostly valley glaciers, while the southern region has a less marked relief with glacial
features close to the geometry of an ice cap (Ke et al., 2015). Surface flow velocities derived from radar measurements spanning
from 2003 to 2011 reveal that nearly 70% of the largest glaciers in the region exhibit a normal flow type, characterized by a
continuous downstream flow. Additionally, 10% of these glaciers are identified as surging glaciers, while the remaining 20%
display nearly stagnant velocity profiles (Yasuda and Furuya, 2013).

In terms of mass balance, West Kunlun is, within the region of High Mountain Asia, affected by what is called the "Karako-
ram anomaly": in 2000-2016, glacier mean elevation change ($0.26 \pm 0.07$ m w e yr$^{-1}$) and region-wide mass balance ($0.14 \pm 0.08$ m w e yr$^{-1}$) was mostly positive (Brun et al., 2017). However, during the 2000-2019 period, Hugonnet et al. (2021)
found a regional mean elevation change rate of -9.6 m yr$^{-1}$ for the Central Asia region (RGI region 13, RGI Consortium,
2017), where West Kunlun is located, indicating an overall downward trend. Specifically, their findings highlight a shift to-
wards thinning, particularly notable in the late 2010s, signifying a potential conclusion to the previously observed Karakoram
anomaly.

Furthermore, the glaciers within the scope of this study are situated in the Tarim Interior River Basin (TIRB), as indicated by
the green shading in Fig. 1a (Lehner et al., 2008). Specifically, the Kunlun mountain range serves as a primary water source for
the Tarim River, a key component of the TIRB, which flows across the Tarim desert (Gao et al., 2010). According to Immerzeel



et al. (2019), the water tower unit of Tarim Interior is defined as the overlap between the Tarim Interior hydrological basin from Lehner et al. (2008) and various mountain ranges from Körner et al. (2017) within the basin. This unit plays a pivotal role in providing water to ecosystems and the downstream population. Tarim is recognized as one of the most significant water units in
Asia, with a notably high contribution of glacier water yield compared to precipitation in the basin. Despite this, the downstream supply often struggles to meet the increasing water demand driven by industrial, domestic, and primarily irrigation needs in the case of TIRB (Immerzeel et al., 2019). Consequently, this basin stands out as one of the most vulnerable, susceptible to the impacts of climate, political, and socioeconomic changes. However, a study focusing on glacier runoff changes using GloGEM (Huss and Hock, 2018) anticipates a rise in Tarim's annual glacier runoff until around 2050, followed by a consistent decline
for the remainder of the 21$^{st}$ century under the RCP4.5 emission scenario.

## 2.2 Ice thickness dataset

In this study, we will compare the timing and magnitude of the peak water predicted using two existing global ice thickness datasets. The first is the consensus for 2019 (abbreviated FARI19, Fig. 1b, Farinotti et al., 2019), which provides a global estimate (except for the Greenland and Antarctic ice sheets) of ice volumes for individual glaciers with the help of five different
models, selected from the Ice Thickness Inter-comparison Project (ITMIX, Farinotti et al., 2017). Inversion methods are based on the use of the principle of mass conservation (Huss and Farinotti, 2012; Maussion et al., 2019), empirical relationships between basal shear stress and glacier elevation change (Linsbauer et al., 2012), or the use of flux thickness inversion (Fürst et al., 2017). One of the common approach in between these models (excepted for Fürst et al., 2017) is the use of glacier by glacier flowline inversions based on elevation data from Digital Elevation Models (DEMs). The second ice thickness model
used in this study (abbreviated MIL22, Fig. 1a, Millan et al., 2022) is based on the inversion using jointly surface ice flow velocity and surface slopes. This inversion makes use of a new global ice velocity product, that provides measurements for 98% of the world's glaciers in the years 2017-2018, at a sampling resolution of 50 meters. Inversions are also based on the SIA, but are performed regionally and in two dimensions. This approach revealed a different picture of the distribution of ice thicknesses and the ice volume of some regions around the Earth (Millan et al., 2022; Hock et al., 2023; Frank and van
Pelt, 2024). Specifically, the Asia region (RGI 13/14/15) displayed a significant difference from the consensus estimate, with total glacier ice volume around 35% higher than the consensus over the same surface area (Farinotti et al., 2019; Millan et al., 2022). The Himalayan region is indeed one of the most uncertain in terms of ice thickness inversion, with very few direct measurements available to constrain the physical parameters of the inversion (e.g., fewer than 10 glaciers were available to calibrate the results of Millan et al., 2022).
In this paper, we are specifically interested in differences over a sub-region of the West Kunlun mountain. Since the consensus model is dated from 2003, and that Millan et al. (2022) uses ice velocity centered in the year 2017-2018, we corrected MIL22 estimates using average glacier mass changes between 2000 and 2020 obtained from DEM differencing (Hugonnet et al., 2021). While solving temporal ambiguities of the thickness models is complicated, since DEM sources and in-situ data are not properly dated, this correction may be a step toward roughly matching the timing of the consensus estimate. Overall,



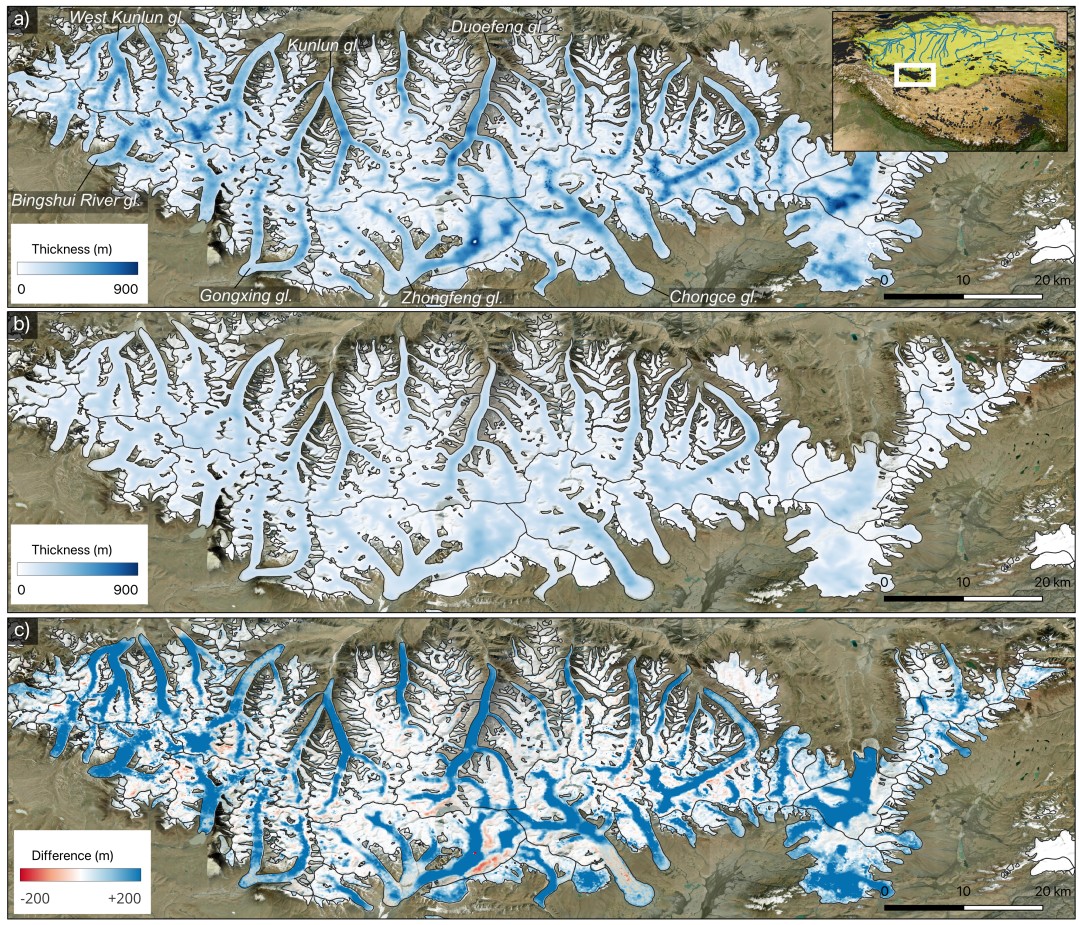

**Figure 1.** Map of the study region: (a) ice thickness from Millan et al. (2022) (with the location of the study set in the Himalayan area); (b) ice thickness from Farinotti et al. (2019); and (c) difference between (a) and ice thickness from Farinotti et al. (2019). Glacier boundaries are from the Randolph Glacier Inventory v6 (RGI Consortium, 2017).

differences after correcting from glacier mass change averages 4.0 km$^3$ from the original MIL22 estimate. Finally, the total ice volume totals 344 km$^3$ and 562 km$^3$ for FARI19 and corrected MIL22 respectively (Fig. 1c).

## 2.3   Description of the glacier model

The Open Global Glacier Model version 1.5 used here is an open-source model that aims to simulate the past and future evolution of glaciers on regional to global scales (https://docs.oggm.org/en/stable/index.html). The model starts by using the
outlines from the RGI (RGI Consortium, 2017) and project them onto a local grid whose resolution depends on glacier size. Topographical data are also added to the grid which sources depends on the glacier location and are part of the NASADEM,



COPDEM, GIMP, TANDEM or MAPZEN. The model computes centerlines with the help of a geometrical routing algorithm (Kienholz et al., 2014) that optimizes the path between local elevation maxima and the glacier's terminus in order to minimize total elevation gain and distance to the glacier terminus. These centerlines are modified into proper flowlines, with grid

points evenly distributed according to a spacing dependent on the grid resolution, and whose elevation is determined from the topography grid. It is worth mentioning that a flowline downstream of the current glacier extent (i.e such as defined by the RGI) is also computed, so that it could possibly grow. For every flowline, a catchment area is determined and used to elaborate geometrical cross-sections at each point, by intersecting the normal to the flowline with the glacier outlines or the catchment areas. These cross-sections are then corrected in respect to the altitude-area distribution of the glacier (Maussion et al., 2019).

After determining these geometrical parameters, OGGM computes the glacier mass balance for each cross-sections, according to a temperature-index model with a single temperature sensitivity factor for the entire glacier (Marzeion et al., 2012; Maussion et al., 2019). The monthly mass balance is calculated as a sum of accumulation and ablation on the glacier, which are functions of temperatures and precipitation. For this purpose, we used gridded climate data, which is by default the latest monthly time series from the Climate Research Unit (CRU, Harris et al., 2020), but also other climate data sources such as

ERA5 (Hersbach et al., 2019) from the European Centre for Medium-Range Weather Forecasts (ECMWF) or climate projections from the Coupled Model Intercomparison Project CMIP5 (Taylor et al., 2012) and CMIP6 (Eyring et al., 2016) for future projections.

Finally, OGGM calibrates parameters of the Surface Mass Balance (SMB) model (such as temperature sensitivity or precipitation factor) with the help of SMB observations from the World Glacier Monitoring Service WGMS (2021). In the latest model

version, OGGM employs global remote sensing data and a new calibration scheme based on global glacier volume changes (Hugonnet et al., 2021). However, this study is based on an earlier calibration framework as described in Maussion et al. (2019), which was the most recent version at the time of the analysis. Because we are performing a sensitivity analysis focused on the role of glacier ice thickness — and all simulations use the same climate forcing and calibrated mass balance parameters — the relative differences in runoff evolution are driven primarily by the initial geometry rather than the specific calibration. This

is consistent with previous studies (Maussion et al., 2019; Huss and Hock, 2015), which emphasize the importance of initial glacier geometry — particularly ice thickness — as a key source of uncertainty in glacier evolution modeling. Thus, the fact that we do not use the updated calibration based on Hugonnet et al. (2021) observations implies that our experiments reflect an average sensitivity of peak water to initial conditions across the region. Implementing the new calibration methods now available in OGGM would allow a closer match to observed mass balance values and consequently more accurate estimates of real

peak water timing and magnitude. However, this is not the aim of the present study that focuses on exploring the sensitivity of peak water timing and magnitude to uncertainties in glacier ice thickness, particularly in a region where discrepancies between inversion methods are substantial.

The model then makes use of the computed surface mass balance to estimate the ice thickness for each cross section. To that aim, OGGM inversion method starts from the mass conservation equation and SIA hypothesis applied to the flowline geometry

in order to derive ice thickness at each cross section as a function of the bedrock shape (rectangular, trapezoidal or parabolic cross section) and "apparent" surface mass balance (equilibrium assumption of the glacier geometry, Maussion et al., 2019),



from which will result the computation of the whole glacier ice thickness and bedrock, used to finally calculate glacier volume, length and area. OGGM uses a dynamical ice flowline forward model to simulate the evolution of the glacier geometry in response to the SMB forcing, resulting itself from any given climate observations or projections.

## 2.4 Integration of ice thickness datasets into OGGM

Due to the fact that Open Global Glacier Model (OGGM) is based on a flowline representation of glacier geometry, limitations can be found when incorporating large two dimensional datasets from remote sensing observations. Consequently, the assimilation of such thickness models into OGGM is not a trivial question. In this paper, we have chosen to investigate the influence of the total ice volume on the glacier contribution to runoff, hence we do not explore spatial and 2D differences between ice thicknesses models. To integrate ice thickness datasets in OGGM, we first calculate, from satellite or model-based observation, the total ice volume for each glacier entity in the region of interest. Secondly, we invert ice thicknesses within the model framework (section 2.3). Finally, the creep parameter $A$ - that describes ice deformation - is calibrated, in order to reach the volume calculated with the observed datasets. If the model cannot converge toward a consistent value of the creep parameter, it will add a non-zero sliding parameter $f_s$ (that is normally set as zero for all glaciers, Maussion et al., 2019). This approach avoids model instabilities that can be found during spin-up processes, with the direct integration of ice thickness dataset in OGGM. Indeed, the latter could potentially disrupt the whole glacier dynamic, since the observed ice thicknesses are not consistent with its modeled volume or its DEM for example, which could lead to the glacier rebalancing itself at the beginning of each simulations.

After the calibration of the OGGM inversions, we find a difference of 0.3 % ± 4.5 % with the volume derived from the ice thickness data calculated from Millan et al. (2022) and Farinotti et al. (2019). This is negligible compared to the difference between the two ice thickness datasets which is roughly 40% (Millan et al., 2022; Farinotti et al., 2019) in the study region. After this assimilation process of the ice thicknesses is done, the glacier initial state is ready to be used for different types of simulations.

In this study, we did not used any spin up procedure because of the large uncertainties in climatic data in Asia. The West Kunlun region is one of the least covered areas by weather stations in Asia, resulting in significant spatial uncertainties in temperature and precipitation data Wester et al. (2019). Hence the climate data used for the spin-up procedure are mostly the results of large interpolations. These interpolations resulted in erroneous positive temperature bias, which reduced the ice volume close to zero for the Farinotti dataset during one of the spin-up tests.

## 2.5 Peak water calculation

We calculate the impact of initial ice thickness on the glacier hydrological mass-balance outputs, and more specifically on the timing and magnitude of the peak water. Considering a fixed glacier area, including glacierized and increasingly non-glacierized terrain, we first derive the annual total runoff as the sum of (1) snow melt on now ice-free area, (2) the ice and seasonal snow melt on glacier, (3) the liquid precipitation on glacier and (4) the liquid precipitation on now ice-free area (Fig. S1). All these variables are outputs of the OGGM model.





While the principle of peak water is often presented as a single maximum value (Huss and Hock, 2015), our simulations often reaches a maximum peak water "plateau", which remain constant for several years or decades. In order to measure the extent of this plateau, we empirically chose to use runoff values included in the top decile. Then, to rule out one single date value for peak water, we selected the median runoff date along the plateau. Similarly for the associated quantity of water runoff, we use the average of the annual runoff on the plateau. After performing simulations on all glaciers with different thicknesses,

we consider the sum of all their annual runoff as the region annual runoff, which is averaged over a 10-years window. It is worth noting that, starting from glacier equilibrium and considering climate warming "enough" high, peak water is thus defined as a tipping point where glacier area has shrunk sufficiently that any further climate evolution can't result in this maximum glacier contribution to basin runoff (Huss and Hock, 2015). In other word, considering a moderate climate warming followed by a temperature decrease, we can reach a temporary maximum of glacier runoff that looks like an "apparent" peak water which is

not a tipping point. In order to investigate the initial ice volume influence on peak water, we design an ensemble of simulations with different values of total ice thickness of all selected glaciers, by multiplying the initial inverted volume from Millan et al. (2022) of the entire glacier by a coefficient ranging from 0.1 to 2 for each individual glacier of the study set. We simulate their evolution using climate data from CRU for historical runs and the GCM MRI-ESM2-0 for future projections (see section 2.6). Similarly, we examined the influence of temperatures on the peak water. To this aim, we simulate the evolution of glaciers with

projection runs (using MIL22 initial ice thicknesses, see section 2.4) by adding temperature biases ranging from -5 to 5°C, meaning that a bias is added to the temperatures series used by the model to calculate surface mass balance (see section 2.3).

## 2.6   Climate data

In order to compute monthly glacier mass balance, we use monthly temperature and precipitation time series as forcings. The simulations include first historical runs, using observational climate data from CRU (such as described in section 2.3).

The results are then used as initial state for projection runs. For these, we used General Circulation Models (GCM) climate data originating from the CMIP6 (Eyring et al., 2016), which employs Shared Socioeconomic Pathways (SSP) as scenario framework (Riahi et al., 2017). In particular, this study was carried out with the climate output from the GCM MRI-ESM2.0 (Yukimoto et al., 2019) regarding three different pathways: SSP1-2.6, SSP5-3.4 and SSP5-8.5. Out of an ensemble of 6 GCMs at our disposition, MRI-ESM2.0 locates itself below the average. Indeed specifically regarding temperatures, the GCM is the

lowest of all, reaching 2°C and 5°C below the average in 2300 under the SSP1-2.6 and the SSP5-8.5 respectively. The choice to use only one GCM would be inaccurate if we were trying to assess accurately and precisely the future evolution of glaciers, but here we are solely interested in the study of the impact of different ice thickness datasets on glaciers and their runoff.





## 3 Results

### 3.1 Peak water sensitivity to initial volume and temperature

Figure 2 presents the sensitivity analysis of the peak water with respect to varying initial thicknesses and temperature bias under SSP1-2.6 and SSP5-8.5 (see section 2.5). It appears clearly that the smaller the initial ice volume of the glaciers, the earlier the peak water occurs, especially under the SSP5-8.5 (Fig. 2a). Indeed, increasing the total ice volume will not significantly advance the timing of the peak water for SSP1-2.6 (~2060), hence suggesting it has not been reached under this climate forcings. On the contrary, multiplying the volume by a factor of two under SSP5-8.5 will delay the timing of the peak water by

roughly 25 years. Similarly, decreasing the initial volume by a factor 0.25, under the same scenario, will advance the timing of the peak water by more than 90 years, from 2150 (multiplying factor of 1) to 2060 (Fig. 2a).

The magnitude of annual runoff at peak water seems to follow a similar trend (Fig. 2b): for SSP1-2.6, it rises slightly to reach a plateau from the coefficient 0.7 and then remains constant, at the level of 2300 Mt of water per year. For small coefficients, annual runoffs at peak water are quite close for both scenarios but then the magnitude of peak water under the

SSP5-8.5 increases more rapidly: halving initial ice volume causes a decrease of 25 % in annual runoff. Then, again, peak water magnitude rises less when we are increasing ice volume.

Temperature bias seems to have little influence on peak water timing under the SSP1-2.6 (Fig. 2c), however it has an impact on annual runoff at peak water: the latter grows perfectly linearly along the range of bias (Fig. 2d). Indeed, a bias of 4°C induces a rise of 30 % in runoff. When it comes to the SSP5-8.5, peak water timing decreases almost linearly as temperature

bias rises; a bias of +2°C induces an advance of more than 15 years of the peak water. Even though annual runoff also increases with temperature bias, its curve is less steep than the one for the SSP1-2.6: a bias of 4°C only results in a rise of 8 % of runoff.

### 3.2 Future projection of evolution using existing ice thickness models

Figure 3 presents glacier simulations using the two global ice thickness estimates, in terms of ice volume, glacierized area, and annual runoff for the time period 2003-2300. We analyze the entire glacier region (i.e. 160 glaciers) by computing the

cumulative evolution of the different variables.

Monthly climate conditions provided by the model MRI-ESM2-0 (averaged over a 20-year window) vary for the different glaciers of the set (Fig. 3c). Depending on the glacier, temperature can vary by almost 4°C and precipitations by nearly 5 kg.m$^{-2}$. Therefore, we used the mean of these variables to compare it with glacier evolution. The two forcing pathways used start differing around 2030 for temperature and around 2075 for precipitation. Regarding temperature, it slowly rises from

lower than -9°C, further continues increasing until 2.5°C around 2215 under the SSP5-8.5, and then stabilizes. Under scenario SSP1-2.6, temperature maintains constant from 2050 at -7.5°C and then decline throughout the decades and returns at an early 21st century level. Precipitation rises from 6 kg m$^{-2}$ in 2015 to 12 kg m$^{-2}$in 2300 for the SSP5-8.5. After a quick gain and loss in the end of the 21$^{st}$ century, precipitation keeps constant from 2125 at nearly 7.5 kg.m$^{-2}$ in SSP1-2.6.

The significant differences between the two ice thickness datasets translates into an equally important one for glaciers ice

volume loss. At the beginning of the 21st century, regional volume calculated from FARI19 (344 km$^3$) represents roughly 60



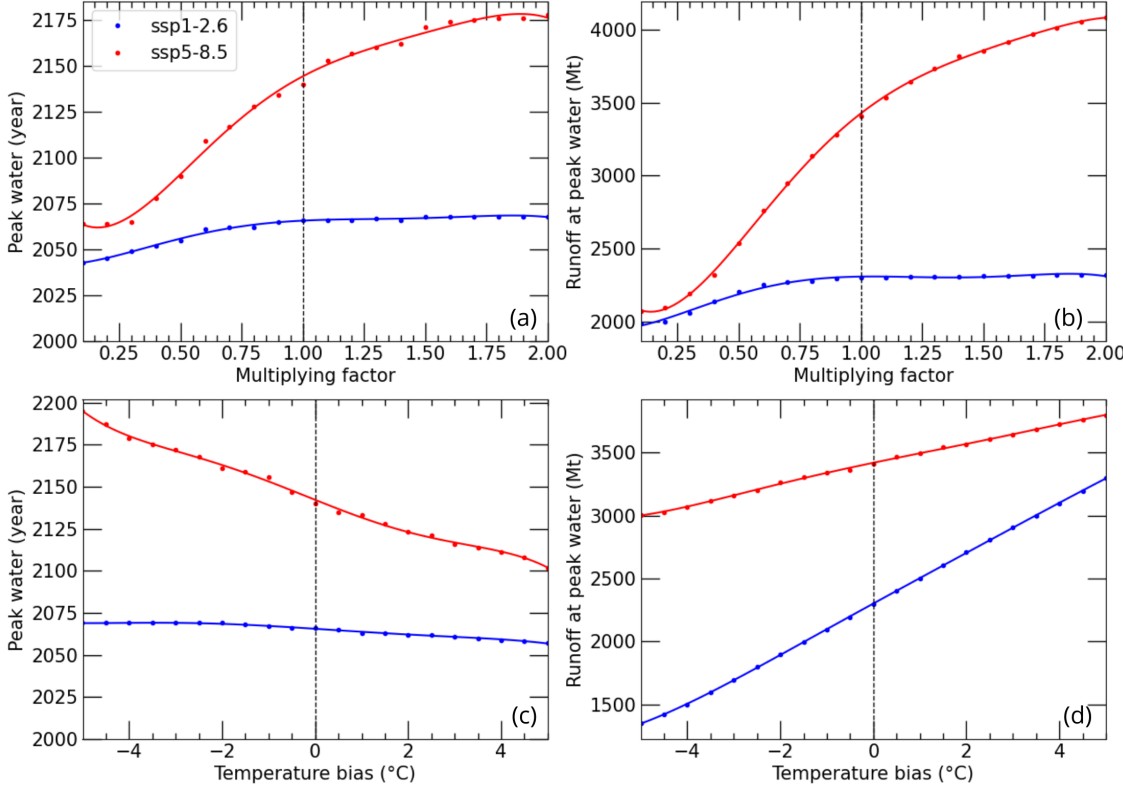

**Figure 2.** Peak water sensitivity: peak water timing **(a)** and runoff at peak water **(b)** for different fractions of the initial ice volume, and peak water timing **(c)** and runoff at peak water **(d)** for different temperature bias. All of these simulations have been carried out using ice thickness data from Millan et al. (2022). Results are presented only for SSP-1.2.6 and SSP-5.8.5, since very little differences are visible between SSP1.2.6 and SSP5.3.4.

% of that computed from MIL22 (562 km$^3$) (Fig. 3a). Under the SSP1-2.6 and the SSP5-3.4, the total volume of ice equals 161 km$^3$ and 26 km$^3$ in 2300 for MIL22 and FARI19 respectively. This translates into a volume reduction of 92 % for FARI19 and 71% for MIL22. Under SSP5-8.5, the volumes declines at a higher rate starting in 2070. In 2300, all the glaciers have almost completely disappeared, with an ice volume of 0.5 km$^3$ and 6 km$^3$ for FARI19 and MIL22 respectively.

As glaciers lose mass, we can also see their surface area receding. The regional glacierized surface areas (2872 km$^2$ at the beginning of simulations) trend start to differ between scenarios around 2050 (Fig. 3b). The glacierized area of the consensus decreases at a faster rate than the one of Millan for the two more optimistic SSPs with a loss of 21 km$^2$.yr$^{-1}$ against 8 km$^2$.yr$^{-1}$ in average for the period 2050-2150. Hence, in 2300 the consensus and the Millan areas have declined by 78 % and 45 % respectively. Curves based on the SSP 5-8.5 still stand out with a higher loss rate, averaging 33 km$^2$.yr$^{-1}$ and 17

km$^2$.yr$^{-1}$ in 2050-2150 for FARI19 and MIL2022 respectively. This leads to a reduced area of 6 km$^2$ for the consensus ice thicknesses and 160 km$^2$ for the MIL2022 dataset, in 2300.




In response to such changes in the glaciers characteristics, their hydrological outputs are also modified along the decades. Here, we present simulated annual runoff, averaged over a 10-years window for better readability (Fig. 3d); for the same reason we also didn't add simulations forced with the SSP5-3.4 on the figure. Runoff starts at the same level for the two datasets, which is 1800 Mt of water per year, equivalent to a runoff of 57 $m^3s^{-1}$, because initial glacier surface area is identical for all simulations. It is worth mentioning that previous work (Gao et al., 2010) estimated average annual glacier runoff in the Tarim River Basin, using observations of annual discharge of mountain river runoff from hydrological stations along with temperature and precipitation monthly time series from national meteorological stations. For the period 1961-2006, the annual runoff was estimated to $144.16 \times 10^8$ $m^3$, i.e., a runoff of 457 $m^3s^{-1}$. The maximum of annual runoff calculated in this study (3400 Mt $yr^{-1}$, Fig. 3d) is roughly equal to a runoff of 108 $m^3s^{-1}$. Hence, while the selected glaciers represent 24 % of the TIRB glacier ice volume (if we use volumes derived from MIL22), our runoff calculation seems to be realistic.

Under the SSP1-2.6, after the first decades of simulation, FARI19 annual runoff goes from 1800 Mt to 2260 Mt in year 2056. The plateau around this peak last approximately 45 years before glacier runoff softly diminishing until reaching a runoff of less than 1000 Mt.$yr^{-1}$ in 2300. MIL22 runoff also rises from 1800 to 2300 Mt.$yr^{-1}$, with peak located in year 2064 and a plateau of the same length, and then declines during the following decades. In year 2300, the runoff value is less than 1500 Mt.$yr^{-1}$. Under the SSP5-8.5, annual runoff increase steadily from 1800 Mt to a peak of 2700 Mt in year 2108 for FARI19. For MIL22, the runoff increases to a peak of 3500 Mt in 2137 for MIL22. In both cases, the plateau lasts roughly 60 years (Fig. 3d). Then the two annual runoff curves decline with an offset in time, and an average runoff difference of 30%. This differences become smaller when they end up almost at same level as their respective SSP1-2.6 annual runoff values in year 2270. The evolution of annual runoff with its four different contributions is also available (Fig. S1).

## 4 Discussion

Our study reveals that the timing and magnitude of peak water are significantly influenced by the initial ice volume of the glaciers under certain conditions. This sensitivity underscores the importance of accurate ice thickness estimations to predict future water availability (Fig. 2). The analysis reveals a particular sensitivity of the study region to the ice thickness model under the SSP5-8.5 scenario, which presents conditions sufficient for glaciers to reach peak water tipping point and toward an ineluctable decrease of their contribution to river runoff (unlike SSP-1.2.6 where an "intermediate" peak in runoff seems to be reached). The difference in the initial ice volume also has a significant influence on the magnitude of peak water (Fig. 2). As shown in Figures 2 and 3, a thicker glacier will at the same time provide more ice for runoff, but will also take longer to melt at lower elevations, keeping ice lower and more out of balance with the climate. This will translate into more negative surface mass balance rates, which in turn will produce increased runoff.

From a methodological point of view, we chose to adjust the creep parameter A to match external ice thicknesses products, which introduces an intrinsic ambiguity: to obtain thicker ice in an inversion, A is often reduced. It is worth noting that a stiffer ice will slows down glacier flow, therefore delaying ice transport toward the ablation area. This has the potential to further postpone the timing of peak water, compounding the delay already induced by the larger total ice volume. Despite this rheolog-



270 ical adjustment, we posit that the timing of peak water is primarily controlled by the initial ice volume and hypsometry. Under the Shallow Ice Approximation, ice flux through a cross section roughly scales as $q \sim Ah^5$ for Glen's flow law with exponent n=3 (assuming slope is unchanged, and a rectangular cross section depending on the thickness) Hutter (1983); Maussion et al. (2019). Therefore, a simple scaling shows that changes in A affect the ice flux linearly, whereas variations in ice thickness have a much stronger, nonlinear impact on the flux. This suggests that initial geometry dominates the glacier response and sets the

275 timing of peak water, while variations in A play a secondary role. Future work could explore more quantitatively the sensitivity of peak water to variations in ice rheology while keeping initial ice thicknesses fixed, in order to better isolate and understand the secondary influence of A on glacier runoff dynamics.

 In the case of the SSP1-2.6 scenario, the changes in the timing of peak water between the two datasets are not particularly significant (Fig. 2 and 3). This can be explained by the impact of the chosen climatic region for the simulations. Indeed, the

280 projected climate under SSP1-2.6 is not sufficient to bring glaciers close to the geomorphological conditions required to reach peak water. Thus, runoff evolution under the optimistic scenarios largely follows the temperature and precipitation trends, which in this case stabilizes after 2050 (Fig. 3c). Therefore, it is possible that, in this specific case, peak water as a tipping point has not actually been reached, and what we observe could more likely be identified as a melting peak. It is worth noting that the glacier volume and area continued to decrease during the entire time period, and that all results concern the average

285 behavior of 160 glaciers.

 The contrasting responses observed between the SSP1-2.6 and SSP5-8.5 scenarios highlight the non-linear influence of temperature bias on glacier runoff and peak water timing. Under SSP1-2.6, although seasonal melting is sufficient to cause continued glacier mass loss, the timing of peak water remains largely unchanged, with only limited variation across the bias range. However, annual runoff at peak water increases almost linearly with temperature bias, suggesting that warming increases

290 surface melt, but without significantly accelerating overall glacier losses. In contrast, under SSP5-8.5, increased temperature bias leads to a more pronounced shift in peak water timing. While runoff also increases with temperature bias, the relative change is smaller than under SSP1-2.6, likely due to the faster depletion of ice volume. This highlights how the influence of temperature uncertainty varies with climate scenario, likely affecting the timing of peak water more under high-emission pathways, and runoff magnitude more under low-emission ones.

295 The improvement of global ice thickness models is a critical issue that depends on several factors. Ice thickness inversion models that rely on surface gradients are only using surface data that carries minimal information about glacier ice dynamics. The inclusion of ice surface velocities, and 2D inversions, introduces a strong constraint into glacier ice thickness inversions, which translates into a much realistic inverted ice thickness field (Millan et al., 2022; Cook et al., 2024). Flow measurements must therefore be continued over time to provide repeated measurements that can be synchronized with other data, such as

300 Digital Elevation Models (DEMs), surface mass balance or penetrating radar measurements (known limitations of the previous method). Future innovative satellite missions could certainly enable the mapping of glacial flow in three dimensions and synchronously with DEMs, which will provide significant advancements in estimating ice volumes, by reducing uncertainties on the temporal mismatch between satellite observations, and improving characterization of basal sliding and ice deformation Kääb et al. (2024).



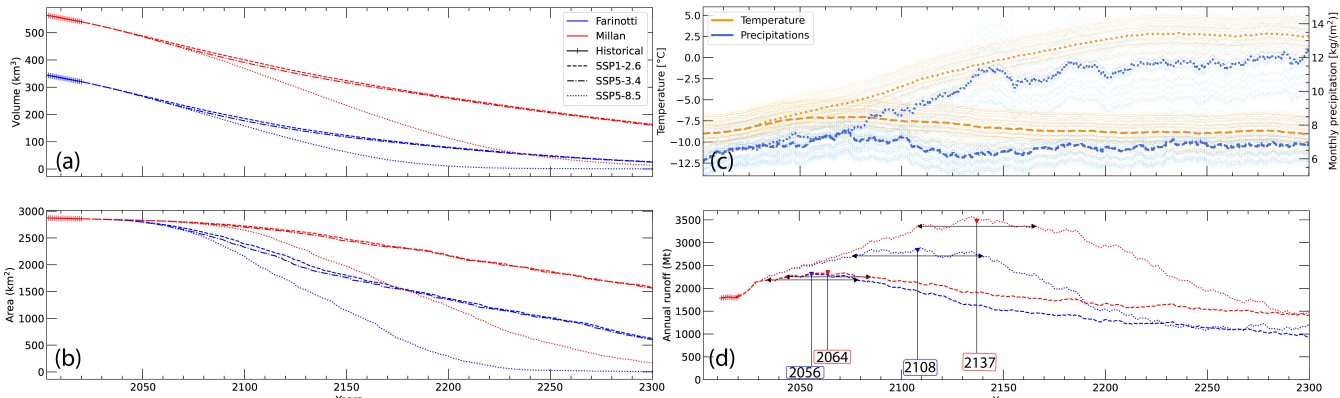

**Figure 3.** Projections of glacier evolution in the region of interest: the cumulative of **(a)** all glacier volumes, of **(b)** glacier areas and of **(d)** annual runoffs of the glaciers of the set, with an assessment of peak water timing, accompanied by **(c)** the temperatures and precipitations projections from GCM MRI-ESM2.0. Time series used for each glaciers of the set are in light, and the respective means for SSPs 1-2.6 and 5-8.5 are in bold.

Additionally, thickness estimates are highly dependent on the calibration of laws describing ice flow, particularly rheology (creep parameter and basal sliding). To calibrate these laws, models use in-situ ice thickness measurements, the spatial scarcity of which leads to significant volume differences, as is the case with glaciers in the high mountains of Asia (Millan et al., 2022; Farinotti et al., 2019). Although advanced new approaches (Bolibar et al., 2023; Cook et al., 2024; Jouvet, 2023) will better constrain these parameters, it is essential to obtain better spatial coverage of in-situ ice thickness in critical regions such as

High Mountain Asia and the Andes. Synchronized planning of measurement campaigns with satellite missions is also crucial to minimize uncertainties related to temporal mismatches between observations, which are subsequently complex to quantify. An open science policy for sharing existing (but unshared) observations, combined with a sufficient environmental approach — prioritizing only essential data collection and simulations — can both reduce model uncertainties and minimize research environmental impact while maximizing societal and scientific value.

Finally, this study highlights the difficulty of accounting for the spatial distribution of ice thicknesses, derived from multi-source inversions, in large-scale glacier models. A major obstacle lies in the challenge of using 2D thickness inversions from external datasets as direct constraints in OGGM, which adjusts the bedrock depth to remain consistent with the simulated glacier dynamics. This critical aspect, which is key to the timing of future glacier evolution—and thus peak water—still remains to be explored. New approaches must be developed to incorporate multi-source thickness measurements as input constraints

in large-scale models. New 2D or 3D models (Jouvet, 2023; Bolibar et al., 2023) have recently emerged and are therefore promising for updating this study. In a broader picture, this study highlights the importance of studying models uncertainty for future glacier changes, especially the initial state of glaciers (Marzeion et al., 2020).



*Code availability.* The code to perform the simulations will be posted on a github repository upon acceptance of the paper.

*Data availability.* All data used in this paper are freely available, and can be accessed at https://www.theia-land.fr/ces-cryosphere/glaciers/,

https://www.research-collection.ethz.ch/handle/20.500.11850/315707 and through the OGGM shop https://docs.oggm.org/en/stable/shop. html.

*Author contributions.* LG, RM and NC conceived and designed the research. LG processed, analyzed data and performed all simulations. All authors participated in the writing of the manuscript.

*Competing interests.* We declare that we have no competing interests.

*Acknowledgements.* LG, NC, RM and JB acknowledge support from the Centre National de la Recherche Scientifique.



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
