# Peer review of "Brief communication: Sensitivity analysis of peak water to ice thickness and temperature: A case study in the Western Kunlun Mountains of the Tibetan Plateau"

_EGUsphere, 2025_

## Referee Comment (RC1)

**Review of Gimenes et al. (2025) 'Sensitivity analysis of peak water to ice thickness and temperature: A case study in the Western Kunlun Mountains of the Tibetan plateau'**

**Summary**
This paper presents an analysis of the runoff contribution, past, present and future, of glaciers in the western Kunlun Shan depending on which of the two currently available global ice-thickness datasets is used to initialise the ice-flow model (OGGM). The authors find substantial differences in the timing of peak water depending on the dataset used, as well as the meteorological forcing used for the future portion of the simulations.

I think this is a well-executed paper that fits the format of a brief communication. The method is solid and the findings well-supported, meaning the paper provides a nice illustration of the importance of model initial conditions in an under-studied region. I have a few minor comments that should be fairly easy to fix, so I recommend the paper be returned to the authors for minor revisions.

Page and line numbers refer to those in the clean version of the revised manuscript.

**Major Comments**
- Language: I confess that at some point on page 7 I got slightly fed up of noting down all the minor grammar points (and even then, I'd let some small ones slide already) and stopped. I hope that the journal's own proof-readers will pick up any remaining ones when it comes to publication, but I'm putting this here just to suggest to the authors that it might be worth having a re-read of the paper before submitting the final version. Mostly, it's all pretty small stuff (a lot of it is singular-plural disagreements between nouns and verbs), but the cumulative effect on some sentences makes them very hard to follow.

**Minor Comments**
- p.1, l.16: 'as other sectors such as'
- p.1, l.17: 'on the timing of peak water'
- p.2, l.29: 'but is not yet applicable at a large scale.' Also, is this true? There have been a couple of recent IGM papers at a regional scale, most recently the Leger et al. (2025) paper for the Alps at the LGM (Leger, T.P.M., Jouvet, G., Kamleitner, S. et al. A data-consistent model of the last glaciation in the Alps achieved with physics-driven AI. Nat Commun 16, 848 (2025). https://doi.org/10.1038/s41467-025-56168-3). By 'large', I assume the authors mean 'global', so I would reformulate this as 'but have not yet been applied at a global scale'.
- p.2, l.33-34: This is a very brief and generic exposition of what the paper is going to do, and also does not at all follow naturally from what went before. There need to be a few more lines here to bridge the gap from the global-scale application of global models talked about in the preceding paragraph to the regional-scale study this paper is doing, and then some more details about exactly what it is this paper is doing (such as the region of concern and why there in particular! Obviously, it's explored in more detail in the following section, but a brief mention here is necessary)
- Introduction generally: Sure, makes sense, but this feels like an introduction to a paper that's doing something on a global scale – everything is couched in terms of, despite their limitations, global-scale glacier models being the only sensible thing to use at a global scale. But this is a regional-scale paper, so there's a bit of a disconnect between the thrust of the introduction and what the paper's actually doing. See my previous comment for one suggestion on how to fix that
- p.2, l.49: 'were mostly positive'
- p.3, l.73: 'One of the common approaches in these models…'
- p.4, l.90: 'correcting for...average...compared to…'
- p.4, l.95: 'projects'
- p.4, l.96: 'whose source...and can be either…'
- p.5, l.105 'cross-section'
- p.5, l.129: 'the OGGM inversion method'
- p.6, l.136: I don't think there's any need to write OGGM out in full again here.

- p.6, l.140: 'ice thickness models'
- p.6, l.145: 'datasets'
- p.6, l.148: 'simulation'
- p.6, l.154: 'use'
- p.6, l.156: The reference isn't properly formatted
- p.7, l.166: 'often reach...which remains...'
- p.7, l.170: 'regional'
- p.7, l.171: I'm not quite sure what the authors mean by '"enough" high' here. I think this sentence might need reworking.
- p.7, Section 2.6: OK, fair enough, the study is only interested in the difference caused by the initial ice thickness, but there needs to be an extra line here to explain why the authors picked that one particular GCM out of the several available.
- Figure 3: It's a long way from where it's referenced in the text. Maybe move it to be a bit closer? It's also rather small – I might suggest having a single column of four panels and thereby allowing each panel to be rather easier to read!

---

## Referee Comment (RC2)

**"Sensitivity analysis of peak water to ice thickness and temperature: A case study in the Western Kunlun Mountains of the Tibetan plateau"**

**Summary**

This study investigates how uncertainties in glacier ice thickness esti██tes and temperature projections affect the timing and magnitude of "peak water" - the maximum glacier runoff that occurs before glaciers shrink sufficiently to reduce water output. Using the Open Global Glacier Model, the authors compare two global ice thickness datasets (Farinotti et al. 2019 and Millan et al. 2022) for 160 glaciers in the Western Kunlun Mountains. They analyze sensitivity to initial ice volume and temperature bias under different climate scenarios. Key findings show that smaller initial ice volumes lead to earlier peak water occurrence, particularly under high-emission scenarios, and that the substantial differences between ice thickness datasets (40% volume difference) translate into important variations in predicted peak water timing and magnitude. Finally, this study highlights the large uncertainty stemming from poorly constrained ice thickness in High Mountain Asia and calls for improved datasets and integration of 2D/3D thickness estimates into models.

The topic is important and the approach is reasonable, but methodological limitations reduce the study's impact. The main finding is valuable, but the paper would benefit from methodological updates, clearer contextualization, and a sharper framing of its contribution. I recommend major revisions.

**Major Comments (not in order of importance)**

1. Introduction. Good background on peak water and ice thickness uncertainty. However, the review of global models (flowline models, Elmer/Ice, emulators) is lengthy and somewhat tangential for a brief communication. I suggest focusing on (1) why the problem matters globally, and (2) regionally, (3) what is uncertain, (4) what the study does to address this. Considering that the study area is already state in the title, maybe it would better to have a few sentences of the "region of interest" here. There are several introduction-like sentences there: "*This basin stands out as one of the most vulnerable, susceptible to the impacts of climate, political, and socioeconomic changes*"

2. Methodological choices. The study acknowledges using an "earlier calibration framework" and justifies this choice based on the research questions. I partly agree, but I think the value of the study lies not only in the comparison between F19 and M22, but also in providing new insights for local ecosystems and communities. If "*the new calibration methods now available in OGGM would allow a closer match to observed mass balance values and consequently more accurate estimates of real peak water timing and magnitude*" then why not use them? Specifically:

   - Using the default climate dataset (CRU) is not a sufficient justification. Given the sensitivity to climate input, I would expect some evidence that this product performs better than other datasets available in OGGM.
   - The argument that "the climate data used for the spin-up procedure are mostly the results of large interpolations" is true for CRU, but this is precisely why ERA5 or W5E5, now the OGGM defaults, would be more appropriate.
   - Stating that "no spin-up procedure was used because of the large uncertainties in climatic data in Asia" is not convincing, since spin-up is the default in OGGM despite large uncertainties being present in most mountain regions.
   - Glacier-specific observations (e.g., geodetic mass balance) could play an important role, as shown in Zekollari et al. (2024). Their use would be especially relevant here, since they are already applied to correct ice thickness.

3. Relying on a single GCM is a recognized weakness, especially given the availability of multiple ready-to-use GCMs in the OGGM Shop. Including an ensemble analysis (e.g., "Out of the 6 GCMs available, MRI-ESM2.0 ranks below the ensemble mean") would substantially strengthen confidence in the results.

4. Considering the almost identical results obtained under SSP126 and SSP534, I recommend carefully double-checking the simulations. Such similarity might indicate a potential issue in the climate forcing data. Since these scenarios differ substantially in their radiative forcing trajectories, one would normally expect more divergence in glacier and runoff responses, especially toward the end of the century.

5. The research question is clear: how sensitive is peak water to uncertainties in initial glacier volume and temperature in a highly glacier-dependent region? While sensitivity analyses are not new, applying them systematically to the Western Kunlun using two global thickness products is a valuable case study. However, the novelty is somewhat limited, as similar experiments have been carried out elsewhere. The contribution could be strengthened by explicitly situating this study within the broader context of global peak water analyses and clarifying what unique insights this regional focus provides. For example, the 40% volume difference between datasets is substantial - discuss whether this represents a common local value or is particularly extreme. From Table 1 in Millan et al. (2022), it seems this would be a very extreme scenario, and therefore some context/discussion is needed.

6. Given the research questions, it is unclear why the authors chose to analyze total glacier runoff rather than glacier melt (snow + ice), since only the latter would be influenced by the choice of ice thickness sources, while precipitation, which is included in the total glacier runoff, remains identical in both scenarios.

**Specific Comments**

**Title and abstract:**
- "Western Kunlun Mountains of the Tibetan plateau" - "plateau" should be capitalized as "Plateau"
- Define SSP in the abstract (first mention)
- "*Temperature bias also significantly influences runoff magnitude and the timing of peak water, especially under high-emission scenarios*" This feels not very informative for the abstract, what is the influence extent? Maybe it would be better to emphasize the regional importance of findings rather than general glacier science.
- Avoid using "*significantly*" if there is not a statistical test applied (is repeated four times in the current version).

**Section 1 (Introduction):**
- "*glacier runoff is expected to rise and reach a maximum*" ... only in certain regions as in many places peak water was already reached.
- "Determining precisely the timing and magnitude" -> "Determining the precise timing"
- The transition between ice thickness products and glacier models could be smoother. Suggest adding 1-2 sentences or split the paragraph according to the main ideas.
- Suggest clarifying why the Western Kunlun was chosen (beyond availability of datasets).

**Section 2 (Methods):**

- Fig1. Indicate the source of the basemap. Please also indicate that the green area corresponds to the Tarim Interior River Basin
- "These are located at very high elevations" -> The glaciers...
- "*mean elevation change rate of -9.6 m yr$^{-1}$*" -> Double check the units
- "Tarim is recognized as one of the most significant water units in Asia, with a notably high contribution of glacier water yield compared to precipitation in the basin". This needs some references, it would be ideally to have some numbers to support this
- "T*his basin stands out as one of the most vulnerable, susceptible to the impacts of climate, political, and socioeconomic changes.*". This should be part of the introduction to justify why this specific region was chosen beyond ice thickness dataset discrepancies (see also major comment on this)
- I don't see the contradiction of having a "however" in the following sentence
- "with the help of five different models, selected from" - should be "using five different models selected from"
- "One of the common approach in between these models" - should be "One common approach among these models"
- "*The first is the consensus for 2019*". This may be misleading as this is year of the study
- "*with a total glacier ice volume around 35% higher than the consensus over the same surface area*", "*The Himalayan region is indeed one of the most uncertain in terms of ice thickness inversion*". Again, this could be used to strengthen the Introduction, and Section 2.2 could be more descriptive.
- "Since the consensus model is dated from 2003". Should not be 2000? As this is the inventory year? Or this region has a different inventory year?
- Good comparison of FARI19 vs MIL22, but the correction for temporal mismatch is only briefly explained. How sensitive are the results to this correction?
- OGGM description: This section reads like a model manual. It could be shortened, focusing on features relevant for this study.
- "The model starts by using the outlines from the RGI" - awkward phrasing
- Using the default climate dataset (CRU) is not a justification and considering the impact of the selection of the climate I would expect some justification indicating that this product is performing better than the alternatives available in OGGM.
- "Due to the fact that Open Global Glacier Model" - wordy, could be "Because OGGM"
- "we first calculate, from satellite or model-based observation" - should be "observations"
- "If the model cannot converge toward a consistent value" - should be "converge on"
- Section 2.5: Consider moving the runoff estimation to the OGGM description section as this is a direct output of OGGM, and not something you would derive from.
- The description of how peak water is determined is unclear. It is not evident whether the calculation considers only the year of maximum runoff or also the duration of the "plateau" around peak water. Typically, the peak water year is identified after applying an 11-year running mean to smooth interannual variability and highlight long-term trends. Please clarify the procedure and consider adding either a supplementary figure or additional explanation in the text.
- L165-182. Paragraph too long. Considering shortening by the main idea.
- 2.6 Climate data: Considering moving the historical data to this section to be more self-contained and also move it after the description of the model to know before hand you will be using future projections.
- The bias correction process of the GCM used in missing in section 2.6.
- I could not find any mention of the spatial resolution of the climate datasets used. Please provide this information. In addition, it only becomes clear in Fig. 3 that the simulations extend until the year 2300—this should be stated explicitly in the Methods section.

**Section 3 (Results):**

- Figure 2 caption: Suggest rephrasing as: "Timing and runoff at peak water for varying ice volume fractions (a–b) and temperature biases (c–d)." Also specify directly in the axis labels that the multiplying factor applies to initial ice volume.
- Figure 3 layout: Consider stacking the panels vertically. The phrase "the glaciers of the set" is unnecessary—if all 160 glaciers in the region are modeled, simply state that.
- Figure 3 variability: The origin of variability in the light lines is unclear. If this reflects glacier altitude or location differences, please clarify; if not relevant, consider removing these lines.
- Sections 3.1–3.2: Both begin with descriptive sentences that duplicate figure captions. These could be removed to improve conciseness.
- Wording corrections:
- "It appears clearly…" → "It is clear that" / "It clearly appears that."
- "Indeed, increasing the total ice volume will not significantly advance" → remove "Indeed."
- "We choosed to adjust" → "We chose to adjust."
- Literature placement: The statement "It is worth mentioning that previous work (Gao et al., 2010)" would fit better in the Discussion.
- Units: Use mm/yr or mm/day for precipitation, and retain m³/s for runoff, to align with hydrological conventions (Fig. 3 and text).
- Supplementary material: Fig. S1 is only mentioned at the end of the Results without discussion; either integrate it into the narrative or remove it.

**Section 4 (Discussion):**

- Having a few subheadings would help to follow the discussion more easily
- "Future work could explore more quantitatively the sensitivity" - should be "explore the sensitivity more quantitatively"
- The contrast between SSP1-2.6 and SSP5-8.5 is well described but somewhat repetitive of the Results section.
- The discussion of satellite missions and data sharing is interesting but feels speculative and less connected to the case study.
- While the study explores sensitivity to ice thickness and temperature, other sources of uncertainty (precipitation projections, model parameters, glacier dynamics, etc) receive limited attention and should be at least mentioned in the discussion.

**Technical and Formatting Issues:**

- Minor errors ("choosed" → "chose"; "did not used" → "did not use").
- Several instances of comma splices
- Inconsistent use of past vs. present tense in methodology sections
- Elimination of wordy constructions and passive voice where possible

---

## Referee Comment (RC3)

**Review of "Brief Communication: Sensitivity analysis of peak water to ice thickness and temperature: A case study in the Western Kunlun Mountains of the Tibetan plateau."**

In this study, Gimenes et al. use OGGM and two ice thickness products to assess the impact of geometric data and temperature bias on the magnitude and timing of peak runoff in the Western Kunlun Mountains. For future projections, the authors employ one GCM simulation extending to 2300 under SSP1-2.6 and SSP5-8.5 scenarios.

The topic is certainly interesting and relevant, but the study provides limited innovation in terms of methodology or analysis. The level of depth is rather modest, and several aspects of the methods and datasets require clarification. Even for a Brief Communication, I believe these points should be in depth addressed. While the results show that initial ice thickness strongly influences runoff, this outcome is somewhat expected. The "so-what" aspect of the study is currently missing and should be emphasised more clearly through deeper analysis and contextualization as well as richer figures. Given the potential of the topic, I recommend major revisions, with the hope that the comments below will help strengthen the manuscript.

**Major comments**

- **Introduction:** The second part of the introduction does not really serve as an introduction to the study itself, but rather as a general overview of glacier modelling approaches. It would be more relevant to provide additional context on which variables most strongly influence glacier runoff and why ice thickness and temperature bias are key factors controlling peak water so that these are chosen. This could then naturally lead into the motivation and significance of the present study. Why is the Western Kunlun Mountains are chosen should also be addressed.

- **Methods:**
  - The authors mention that the ice thickness data from Millan et al. (2022) were corrected to match the consensus estimate (reported for 2003), but the procedure is not entirely clear. Was the ice thickness change from 2000–2019 (Hugonnet et al., 2021) applied across the full domain, or only partially (e.g. 2003–2017)? It would also be helpful to clarify what the reported average value of 4.0 km³ (line 90) represents.
  - The calibration section needs revision. It currently reads as a mix of "limitations" and "comparison with previous studies", which weakens the message and the perceived robustness of the study. Why not rerunning OGGM using the new configuration to ensure consistency with the latest developments?
  - The discussion of the temperature bias (lines 180-181) is too brief, as it is a central element of the title. Is this bias the same as that derived during the calibration? Was the same procedure repeated but with a uniform bias applied over the full simulation period? Please clarify this.

- **Results and discussion:**
  - Figures 2 and 3 could be further optimised. Axis limits are inconsistent, which makes comparison difficult. Colour choices are sometimes awkward; using the IPCC standard colour palette would improve clarity. Labels are not always self-explanatory, making the caption crucial for interpretation, and overall label thickness should be increased for readability.
  - The discussion lacks comparison with previous studies or observational runoff datasets that could provide at least partial validation. As a result, the broader context

and implications of the results remain unclear.

**Minor comments and suggestions:**

**Abstract**

- Line 1: Sensitivity to what? Please specify (-> ice thickness and temperature?).
- Line 4: Replace Predicted with Projected.

**Introduction**

- Line 11: Add a reference for the Sea Level Rise component.
- Line 12: Remove also.

**Data and Methods**

- Line 37: Replace with "more precisely, the Western Kunlun Mountains." (or West?)
- Line 38: Clarify why RGI v6.0 is used to define the study area (maybe better add this reference in the next sentence).
- Line 45: The phrase "these glaciers" currently implies 10% of 70%. Please reword for clarity.
- Line 50: A thinning rate of $-9.6$ m yr$^{-1}$ seems unrealistically high? Please verify the source or units.
- Region 13 is generally thinning, but the Karakoram is an exception within RGI13, what does this imply for interpreting the Hugonnet (2021) regional value? Why not taking the Karakoram region from Hugonnet (2021) data separately?
- Figure 1: The inset zoom on the study area in HMA is too small. Please revise and consider adding the ice thickness dataset already in the panel (not only in the caption). The figure should be interpretable without relying on the caption. If ice thickness measurements exist in the study region, they could be included.
- Line 63: Add a reference for this statement. This could also be moved to the introduction to justify the regional focus.
- Line 67: Replace Predicted with Projected or Simulated.
- Line 68: The consensus (or community) estimate was published in 2019 but represents 2003 (I thought actually more 2000), not 2019.
- Line 73: Clarify the phrase "the use of glacier by."
- Line 84: Confirm whether Farinotti et al. (2019) used measurements.
- Lines 84-89: The correction method for the Millan dataset is unclear — was the Hugonnet thickness change simply added?
- The OGGM initialization description is too detailed for a Brief Communication. Focus on "We use OGGM, a model…" and emphasize what was new or modified for this study.
- Lines 108–112: Clarify which dataset was ultimately used and for what time period.
- Lines 113–127: This section reads more like discussion than methods. It could be shortened substantially. Based on line 121, it even seems that part of the research question has already been answered.
- Lines 154–159: This belongs more logically in the description of the reanalysis dataset. Ensure consistent verb tense (past or present) throughout each section.

- Consider first presenting the individual glacier runoff results before aggregating to the regional scale.
- There is some mixing of climate data, geometry, and model description; please reorder for clarity.
- Why was only one GCM used, given that six are available? The paper itself acknowledges that using a single GCM limits accuracy — please justify this choice.

**Results**

- Figure 2: It is unclear what the "multiplication factor" represents. Also, clarify the meaning of "temperature bias." Ensure consistent y-axis ranges across panels and apply IPCC colours for SSP1-2.6 and SSP5-8.5.
- Figure 3: Move the figure closer to the relevant text. The colour scheme used to distinguish Farinotti et al. (2019) and Millan et al. (2022) is identical to that used for SSP scenarios in Figure 2 — please revise for clarity. Label font size should be increased.
- The content of panel C is not intuitive, why not show annual mean temperature and total precipitation instead?
- Lines 241–246: Move to the discussion section.

**Discussion**

- Consider subdividing this section into subsections for clarity.
- Is Millan et al. (2022) systematically thicker across the entire glacier, or mainly in the accumulation area? If the latter, would that difference significantly affect runoff?
- Line 280: Please clarify what "sufficient" refers to.
- Lines 297-304: The outlook is vague. The discussion should place the results in a broader context and compare them with previous studies. Many results are rather intuitive (e.g. thicker ice leads to delayed and higher peak water). It would be more appropriate to revisit the research questions rather than expand on new remote-sensing products, which remain uncertain if and when they will come.
- Why is there no **conclusion section**? Please include one to summarise the key findings and implications, even in a couple of sentences.

---

## Author Comment (AC1)

**Reviewer 1**

**Summary**

This paper presents an analysis of the runoff contribution, past, present and future, of glaciers in the western Kunlun Shan depending on which of the two currently available global ice-thickness datasets is used to initialise the ice-flow model (OGGM). The authors find substantial differences in the timing of peak water depending on the dataset used, as well as the meteorological forcing used for the future portion of the simulations. I think this is a well-executed paper that fits the format of a brief communication. The method is solid and the findings well-supported, meaning the paper provides a nice illustration of the importance of model initial conditions in an under-studied region. I have a few minor comments that should be fairly easy to fix, so I recommend the paper be returned to the authors for minor revisions.

Page and line numbers refer to those in the clean version of the revised manuscript.

Authors: We would like to thank Reviewer 1 for the constructive overall comments on our study. We provide below a detailed response to each comments.

**Major Comments**

• Language: I confess that at some point on page 7 I got slightly fed up of noting down all the minor grammar points (and even then, I'd let some small ones slide already) and stopped. I hope that the journal's own proof-readers will pick up any remaining ones when it comes to publication, but I'm putting this here just to suggest to the authors that it might be worth having a re-read of the paper before submitting the final version. Mostly, it's all pretty small stuff (a lot of it is singular-plural disagreements between nouns and verbs), but the cumulative effect on some sentences makes them very hard to follow.

Authors: We are sorry for the inconvenience caused by all grammar errors and we tried to correct all of these in the revised version.

**Minor Comments**

• p.1, l.16: 'as other sectors such as'

Authors: Done.

• p.1, l.17: 'on the timing of peak water'

Authors: Done, with the sentence being changed to "Inversion of ice thicknesses is a major source of uncertainty, influencing modeled ice volumes and consequently the timing of peak water" at L19-20.

• p.2, I.29: 'but is not yet applicable at a large scale.' Also, is this true? There have been a couple of recent IGM papers at a regional scale, most recently the Leger et al. (2025) paper for the Alps at the LGM (Leger, T.P.M., Jouvet, G., Kamleitner, S. et al. A data-consistent model of the last glaciation in the Alps achieved with physics-driven Al. Nat Commun 16, 848 (2025). https://doi.org/10.1038/s41467-025-56168-3). By 'large', I assume the authors mean 'global', so I would reformulate this as 'but have not yet been applied at a global scale'.

Authors: We changed the introduction section (see response to Reviewer #2's major comment) and therefore removed the model review part.

• p.2, I.33-34: This is a very brief and generic exposition of what the paper is going to do, and also does not at all follow naturally from what went before. There need to be a few more lines here to bridge the gap from the global-scale application of global models talked about in the preceding paragraph to the regional-scale

study this paper is doing, and then some more details about exactly what it is this paper is doing (such as the region of concern and why there in particular! Obviously, it's explored in more detail in the following section, but a brief mention here is necessary)

Authors: Agreed. We have added a more comprehensive paragraph at L32-38, detailing the region and the framework of the study:

"In this study, we examine how uncertainties in ice thickness estimates and temperature affect the timing, magnitude, and duration of peak water in a region with strong variability between existing datasets and strong vulnerability to climate change in terms of water supply (Immerzeel et al., 2020): the Western Kunlun Mountains on the Tibetan Plateau. We first propose a methodology to assimilate ice thickness inversions into the OGGM model, with an approach that can be applied globally. We then perform sensitivity experiments by perturbing both initial ice volume and climate forcing to quantify their combined effects on peak water. Finally, we illustrate the role of glacier geometry by comparing two widely used ice thickness datasets that differ substantially in this region."

• Introduction generally: Sure, makes sense, but this feels like an introduction to a paper that's doing something on a global scale – everything is couched in terms of, despite their limitations, global-scale glacier models being the only sensible thing to use at a global scale. But this is a regional-scale paper, so there's a bit of a disconnect between the thrust of the introduction and what the paper's actually doing. See my previous comment for one suggestion on how to fix that

Authors: Agreed. Considering a similar comments from Reviewer #2 we have removed the description on global vs regional models, and are now focusing on the following points in the introduction: (1) why the problem matters globally, and (2) regionally, (3) what is uncertain, (4) what the study does to address this.

• p.2, I.49: 'were mostly positive'

Authors: Done.

• p.3, l.73: 'One of the common approaches in these models...'

Authors: Done.

• p.4, I.90: 'correcting for...average...compared to...'

Authors: Done, sentence was changed to "differences after correction for glacier mass change average 6 km3 compared to the original MIL22 estimate".

• p.4, l.95: 'projects'

Authors: This specific sentence was removed from the revised manuscript.

• p.4, I.96: 'whose source...and can be either...'

Authors: The entire phrasing was changed to "topographical data from various sources (NASADEM, COPDEM, GIMP, TANDEM or MAPZEN depending on the glacier location)" at L101-102.

• p.5, I.105 'cross-section'

Authors: This specific sentence was removed from the revised manuscript.

• p.5, I.129: 'the OGGM inversion method'

Authors: The entire phrasing was changed to "OGGM uses a flux-based ice flow model to solve a mass conservation equation along flowlines, under the SIA hypothesis, deriving ice thickness at each cross section." at L111-112.

• p.6, I.136: I don't think there's any need to write OGGM out in full again here.

Authors: Done.

• p.6, I.140: 'ice thickness models'

Authors: Done.

p.6, I.145: 'datasets'

Authors: Done.

• p.6, I.148: 'simulation'

Authors: Done.

• p.6, l.154: 'use'

Authors: This paragraph was removed in the revised version in accordance with Reviewer #2's comments on the model framework used.

• p.6, I.156: The reference isn't properly formatted

Authors: See above response.

• p.7, l.166: 'often reach...which remains...'

Authors: Done.

• p.7, l.170: 'regional'

Authors: Done at L156.

• p.7, I.171: I'm not quite sure what the authors mean by "enough" high' here. I think this sentence might need reworking.

Authors: Agreed. We have rephrased the entire sentence at L163-165 to :

"It is worth noting that, starting from glacier equilibrium and considering a climate that has warmed enough to cause substantial glacier retreat, peak water represents the tipping point beyond which any additional warming leads to a decline in glacier contribution to basin runoff (Huss and Hock, 2015)."

• p.7, Section 2.6: OK, fair enough, the study is only interested in the difference caused by the initial ice thickness, but there needs to be an extra line here to explain why the authors picked that one particular GCM out of the several available.

Authors: Agreed. In the revised version we now use 5 different GCMs. Out of the 5 GCMs available in OGGM that extend until 2300, one (CanESM5) was put aside because presenting very sharp temperature increase in the region (see figure below).

Temperature and precipitation in the region under GCM CanESM5 under SSP1-2.6 (blue), SSP5-3.4-OS (yellow) and SSP5-8.5 (red).

• Figure 3: It's a long way from where it's referenced in the text. Maybe move it to be a bit closer? It's also rather small – I might suggest having a single column of four panels and thereby allowing each panel to be rather easier to read!

Authors: Done. Figure 3 was moved to section 3.2 ("Future projection of evolution using existing ice thickess models"). We have also modified the figure accordingly into a single column of four panels. Considering that we now use 5 GCMs we included also included error bars in Figure 3.

---

## Author Comment (AC2)

**Reviewer 2**

**Summary**

This study investigates how uncertainties in glacier ice thickness estima-tes and temperature projections affect the timing and magnitude of "peak water" - the maximum glacier runoff that occurs before glaciers shrink sufficiently to reduce water output. Using the Open Global Glacier Model, the authors compare two global ice thickness datasets (Farinotti et al. 2019 and Millan et al. 2022) for 160 glaciers in the Western Kunlun Mountains. They analyze sensitivity to initial ice volume and temperature bias under different climate scenarios. Key findings show that smaller initial ice volumes lead to earlier peak water occurrence, particularly under high-emission scenarios, and that the substantial differences between ice thickness datasets (40% volume difference) translate into important variations in predicted peak water timing and magnitude. Finally, this study highlights the large uncertainty stemming from poorly constrained ice thickness in High Mountain Asia and calls for improved datasets and integration of 2D/3D thickness estimates into models.

The topic is important and the approach is reasonable, but methodological limitations reduce the study's impact. The main finding is valuable, but the paper would benefit from methodological updates, clearer contextualization, and a sharper framing of its contribution. I recommend major revisions.

Authors: We would like to thank Reviewer 2 for the thorough and detailed comments that contributed in largely improving the paper quality and robustness.

**Major Comments (not in order of importance)**

1. Introduction. Good background on peak water and ice thickness uncertainty. However, the review of global models (flowline models, Elmer/Ice, emulators) is lengthy and somewhat tangential for a brief communication. I suggest focusing on (1) why the problem matters globally, and (2) regionally, (3) what is uncertain, (4) what the study does to address this. Considering that the study area is already state in the title, maybe it would better to have a few sentences of the "region of interest" here. There are several introduction-like sentences there: "This basin stands out as one of the most vulnerable, susceptible to the impacts of climate, political, and socioeconomic changes"

Authors: Agreed. We have now completely reshaped the introduction (L10-38), removing the review section on models, and focusing on (1) why the problem matters globally, and (2) regionally, (3) what is uncertain, (4) what the study does to address this.

- 2. Methodological choices. The study acknowledges using an "earlier calibration framework" and justifies this choice based on the research questions. I partly agree, but I think the value of the study lies not only in the comparison between F19 and M22, but also in providing new insights for local ecosystems and communities. If "the new calibration methods now available in OGGM would allow a closer match to observed mass balance values and consequently more accurate estimates of real peak water timing and magnitude" then why not use them? Specifically:
  - Using the default climate dataset (CRU) is not a sufficient justification. Given the sensitivity
    to climate input, I would expect some evidence that this product performs better than other
    datasets available in OGGM.

Authors: Agreed. We are now using the W5E5 dataset in the revised version since it is used as baseline climate in OGGM. Additionally W5E5 is employed as input for the bias-correction of ISIMIP3b CMIP6 GCMs, therefore it seems relevant for us to use it for bias-correction of the CMIP6 GCMs that we handled in this study.

 The argument that "the climate data used for the spin-up procedure are mostly the results of large interpolations" is true for CRU, but this is precisely why ERA5 or W5E5, now the OGGM defaults, would be more appropriate.

Authors: Agreed. We are now using W5E5 dataset for spin-up procedures.

 Stating that "no spin-up procedure was used because of the large uncertainties in climatic data in Asia" is not convincing, since spin-up is the default in OGGM despite large uncertainties being present in most mountain regions.

Authors: Agreed. We used the dynamical spin-up available in OGGM in all simulations, except for certain sensitivity tests. For the sensitivity experiments on initial ice volume, the prescribed volumes were much lower or higher than realistic values, preventing the iterative dynamical spin-up from converging to match the RGI glacier area given the resulting glacier geometry and the calibrated mass balance model. Similarly, for comparison purposes, the dynamical spin-up was not applied in the temperature bias sensitivity tests. Instead of the dynamical spin-up we have conducted historical simulations usg W5E5 data from 2000 until 2020 in order to initialize glacier state before future projections simulations. The text was modified accordingly at L112- 115 and L169-171 to describe these changes.

"We also make use of a feature from the latest version of OGGM which is a dynamic spin-up that can provide glacier initial state for the year 2020 by reconstructing its recent past while ensuring that modeled glacier area and observed one are matched within 1 % at the inventory date under historical climate forcing (Aguayo et al., 2023; Zekollari et al., 2024)."

"We do not use the dynamical spin-up in this set up since it can not converge to match the RGI area with the amount of simulated reduced or increased ice volume. Instead, after ice thicknesses assimilation we simulate glaciers evolution from 2000 to 2020 simply using historical W5E5 data to initialize glaciers before future projections (see section 2.4)."

 Glacier-specific observations (e.g., geodetic mass balance) could play an important role, as shown in Zekollari et al. (2024). Their use would be especially relevant here, since they are already applied to correct ice thickness.

Authors: Agreed. We now use the mass balance calibration on geodetic mass balance observations in the revised version. A description was added at L108-110 :

"The surface mass balance model is calibrated on geodetic mass balance observations obtained from remote sensing (Hugonnet et al., 2021) for the 2000-2019 time period, using the W5E5v2.0 climate dataset as forcing (Lange et al., 2021).

3. Relying on a single GCM is a recognized weakness, especially given the availability of multiple ready-to-use GCMs in the OGGM Shop. Including an ensemble analysis (e.g., "Out of the 6 GCMs available, MRI-ESM2.0 ranks below the ensemble mean") would substantially strengthen confidence in the results.

Authors: Agreed. This was modified accordingly. See earlier response to Reviewer #1.

4. Considering the almost identical results obtained under SSP126 and SSP534, I recommend carefully double-checking the simulations. Such similarity might indicate a potential issue in the climate forcing data. Since these scenarios differ substantially in their radiative forcing trajectories, one would normally expect more divergence in glacier and runoff responses, especially toward the end of the century.

Authors: Results were updated using the new modeling framework (see previous comments). We are therefore able to see more differences in changes in volume and area of glacier between SSP126 and 534. We also added in the paper that the latter is actually the SSP5-3.4 Overshoot (OS) scenario and that is is "actually not an intermediate scenario since it explores the implications of a peak and decline in forcing during the 21st century (Lee et al., 2021)" at L131-132. It is described the following way in the AR6 of IPCC: "By 2300, warming under the SSP5-3.4-OS scenario decreases from a peak around year 2060 to a level very similar to SSP1-2.6.". This is consistent with climate evolution represented in Fig. 3d of the revised manuscript. However, since the two scenario are still quite close, we only display SSP126 and SSP585 for the peak water in order to enhance readability, but Fig. 3c is available with SSP5-3.4 in the supplementary. Figure 3 was updated accordingly. We chose those scenarios since they are only ones available in OGGM running until the year 2300.

5. The research question is clear: how sensitive is peak water to uncertainties in initial glacier volume and temperature in a highly glacier-dependent region? While sensitivity analyses are not new, applying them systematically to the Western Kunlun using two global thickness products is a valuable case study. However, the novelty is somewhat limited, as similar experiments have been carried out elsewhere. The contribution could be strengthened by explicitly situating this study within the broader context of global peak water analyses and clarifying what unique insights this regional focus provides. For example, the 40% volume difference between datasets is substantial - discuss whether this represents a common local value or is particularly extreme. From Table 1 in Millan et al. (2022), it seems this would be a very extreme scenario, and therefore some context/discussion is needed.

Authors: Agreed. We have added in the introduction section a sentence situating this study within a broader context of other peak water studies and clarifying what new insights are brought here (L28-31):

"While several studies have investigated the estimation of peak water at regional scales \citep{pwhuss,caro}, there are, to date, few quantitative assessments of how uncertainties in initial glacier geometry and climate model biases influence its timing and magnitude."

Furthermore, along with the reshaping of the introduction, we are now specifically stating that this mountain range was chosen because of the extreme difference in ice thickness models (L27-28).

Given the research questions, it is unclear why the authors chose to analyze total glacier runoff rather than glacier melt (snow + ice), since only the latter would be influenced by the choice of ice thickness sources, while precipitation, which is included in the total glacier runoff, remains identical in both scenarios.

Authors: We thank the reviewer for this comment. Indeed, since precipitations are the same across simulations, any difference in total glacier runoff necessarily reflects the impact of ice thickness and geometry on melt water production, storage, and release. Furthermore, our analysis focuses on total glacier runoff rather than melt alone because our objective is to quantify the hydrological implications of differences in initial ice thickness. Indeed, total runoff better represents what downstream hydrological systems experience. Furthermore, in the temperature bias sensitivity experiments, only air temperature was perturbed, while precipitation remained identical across runs. Nevertheless, changing temperature could potentially affect the partitioning between solid and liquid precipitation, thereby impacting seasonal snow accumulation and melt timing. Even though precipitation remains constant, this shift in phase and timing could modify the runoff regime and consequently the timing and magnitude of the peak water. Additionally, we find it more convenient to have results in terms of total glacier runoff for potential comparison in future studies.

**Specific Comments Title and abstract:**

• "Western Kunlun Mountains of the Tibetan plateau" - "plateau" should be capitalized as "Plateau"

Authors: Done.

• Define SSP in the abstract (first mention)

Authors: Done.

• "Temperature bias also significantly influences runoff magnitude and the timing of peak water, especially under high-emission scenarios" This feels not very informative for the abstract, what is the influence extent? Maybe it would be better to emphasize the regional importance of findings rather than general glacier science.

Authors: We have modified the abstract accordingly, to better provide the influence extent as follow: "Temperature bias also notably influences the timing runoff by delaying peak water timing in the region by roughly 13 years for each bias degree".

• Avoid using "significantly" if there is not a statistical test applied (is repeated four times in the current version).

Authors: The word was removed from the abstract and replaced by "notably".

**Section 1 (Introduction):**

• "glacier runoff is expected to rise and reach a maximum" ... only in certain regions as in many places peak water was already reached.

Authors: Done, we added the following phrase (L13-14) to clarify that: "(if it is not already reached as it is the case in many regions)"

"Determining precisely the timing and magnitude" -> "Determining the precise timing"

Authors: Done.

- The transition between ice thickness products and glacier models could be smoother. Suggest adding
- 1-2 sentences or split the paragraph according to the main ideas.

Authors: Agreed, introduction has been divided into four paragraphs and largely reshaped, see earlier response to major comment on the introduction.

Suggest clarifying why the Western Kunlun was chosen (beyond availability of datasets).

Authors: Agreed, the choice of the region of interest is now explicitly in the introduction and addressed in a paragraph (L25-31):

"The Western Kunlun Mountains are located at the norther edge of the Tibetan Plateau. This is a major glacierized region within the Tarim Interior River Basin (TIRB), where glacial meltwater provides contribution to downstream water resources (Immerzeel et al., 2020). Despite its importance, this region is subject to the largest discrepancies among existing ice thickness datasets, making it an ideal case to illustrate how geometry uncertainty influences modeled glacier runoff. While several studies have investigated the estimation of peak water at regional and global scales (Huss and Hock, 2018; Caro et al., 2025), there are to date, few quantitative assessments of how uncertainties in initial glacier geometry and climate model biases influence its timing and magnitude."

**Section 2 (Methods):**

• Fig1. Indicate the source of the basemap. Please also indicate that the green area corresponds to the Tarim Interior River Basin

Authors: We added this indications in the caption of Figure 1: "Basemap is a mosaic of images from Copernicus Sentinel-2 data generated via sentinel-hub (Sinergise Solutions d.o.o.). Green area on the insert map corresponds to the Tarim Interior River Basin."

"These are located at very high elevations" -> The glaciers...

Authors: Done.

"mean elevation change rate of -9.6 m yr-1" -> Double check the units

Authors: We double checked this value and realized we had read the wrong rate. Mean elevation change rate is now of  $-0.23 \pm 0.04$  m yr-1.

• "Tarim is recognized as one of the most significant water units in Asia, with a notably high contribution of glacier water yield compared to precipitation in the basin". This needs some references, it would be ideally to have some numbers to support this

Authors: We added the reference of (*Immerzeel et al., 2020*) at the end of the sentence. Glacier and precipitation contributions are expressed in the referenced paper as components of a supply index, therefore we think it will not be very obvious to the reader to add it in the paper.

• "This basin stands out as one of the most vulnerable, susceptible to the impacts of climate, political, and socioeconomic changes.". This should be part of the introduction to justify why this specific region was chosen beyond ice thickness dataset discrepancies (see also major comment on this)

Authors: Agreed, we addressed that in previous comments on introduction.

• I don't see the contradiction of having a "however" in the following sentence

Authors: Agreed, we removed "however" from the following sentence at L67-69.

• "with the help of five different models, selected from" - should be "using five different models selected

from"

Authors: Done.

• "One of the common approach in between these models" - should be "One common approach among these models"

Authors: Done.

"The first is the consensus for 2019". This may be misleading as this is year of the study

Authors: Agreed, this was clarified with the phrasing "The first is the consensus obtained in 2019" at L72.

• "with a total glacier ice volume around 35% higher than the consensus over the same surface area", "The Himalayan region is indeed one of the most uncertain in terms of ice thickness inversion". Again, this could be used to strengthen the Introduction, and Section 2.2 could be more descriptive.

Authors: The first sentence mentioned in the commentary was indeed removed from section 2.2 and added to the introduction as "Yet large discrepancies persist between these datasets, particularly in High Mountain Asia, where total ice volume estimates differ by up to 35% between products (Farinotti et al., 2019; Millan et al., 2022)." at L22-24. We think that "The Himalayan region is indeed one of the most uncertain in terms of ice thickness inversion" should remain in the section 2.2 to further mention the lack of thickness measurements available to constrain the datasets.

• "Since the consensus model is dated from 2003". Should not be 2000? As this is the inventory year? Or this region has a different inventory year?

Authors: we changed the phrasing to "Since the consensus model is roughly dated from 2000" at L89-90. However we also mentioned (L90-91) that the inventory year is 2010 in this region, but the DEM used for thickness inversion (SRTM) is from 2000-2001, so we deemed FARI19 thicknesses to be centered indeed on the year 2000.

• Good comparison of FARI19 vs MIL22, but the correction for temporal mismatch is only briefly explained. How sensitive are the results to this correction?

Authors: We clarified the procedure for the correction and its impact on MIL22 ice volume following sentences at L91-97:

"we corrected MIL22 estimates to get corresponding thicknesses in the year 2000. As correction, we simply subtracted average glacier thickness changes of Hugonnet et al. (2021) between 2000 and 2019, which is obtained from DEM differencing, to MIL22 thicknesses. While solving temporal ambiguities of the thickness models is complicated, since DEM sources and in-situ data are not properly dated, this correction may be a step toward roughly matching the timing of the consensus estimate. Overall, differences after correction for glacier mass change average 6 km3 compared to the original MIL22 estimate, representing 1 % of the latter."

• OGGM description: This section reads like a model manual. It could be shortened, focusing on features relevant for this study.

Authors: We significantly shortened the section with the new modeling framework (see responses to major comments).

"The model starts by using the outlines from the RGI" - awkward phrasing

Authors: We changed the phrasing to "OGGM is an open-source glacier evolution model that uses RGI outlines" at L101.

• Using the default climate dataset (CRU) is not a justification and considering the impact of the selection of the climate I would expect some justification indicating that this product is performing better than the alternatives available in OGGM.

Authors: see previous response to major comment on climate dataset.

"Due to the fact that Open Global Glacier Model" - wordy, could be "Because OGGM"

Authors: agreed, changed to "Since OGGM".

• "we first calculate, from satellite or model-based observation" - should be "observations"

Authors: Done.

"If the model cannot converge toward a consistent value" - should be "converge on"

Authors: Done.

• Section 2.5: Consider moving the runoff estimation to the OGGM description section as this is a direct output of OGGM, and not something you would derive from.

Authors: Agreed, we move the part "Indeed, considering a fixed glacier area including glacierized and increasingly non-glacierized terrain, the annual total runoff computed in OGGM is derived as the sum of snow melt on now ice-free area, the ice and seasonal snow melt on glacier, the liquid precipitation on glacier, and the liquid precipitation on now ice-free area (Fig. S1)." to the end of the OGGM section, at L116-118.

• The description of how peak water is determined is unclear. It is not evident whether the calculation considers only the year of maximum runoff or also the duration of the "plateau" around peak water. Typically, the peak water year is identified after applying an 11-year running mean to smooth inter-annual variability and highlight long-term trends. Please clarify the procedure and consider adding either a supplementary figure or additional explanation in the text.

Authors: Agreed, we added clarifications on the calculation at L157-162, to explain that it also considers the duration of the plateau: "While the principle of peak water is often presented as a single maximum value (Huss and Hock, 2015), our simulations often reach a maximum regional runoff "plateau", which remains constant for several years or decades. To measure the extent of this plateau, we empirically chose to define it as the top 10 % of simulated runoff values for SSP5-8.5 (and as the top 5 % for other SSPs). To pick one single date value for peak water timing, we selected the median date of the plateau's temporal extent. Similarly for the associated quantity of water runoff, we use the average of the plateau's values.."

• L165-182. Paragraph too long. Considering shortening by the main idea.

Authors: We tried to shortened the paragraph in the revised version, but this is kind of contradictory with previous comment.

• 2.6 Climate data: Considering moving the historical data to this section to be more self-contained and also move it after the description of the model to know before hand you will be using future projections.

Authors: Agreed, climate data section was moved right after OGGM section and historical data was added in it.

• The bias correction process of the GCM used in missing in section 2.6.

Authors: Agreed, we added mention of the bias correction at L127-128:

"The reanalysis data from 2000-2019 is used as reference climatology for bias correction of the 5 GCMs, following the anomaly method implemented in OGGM".

• I could not find any mention of the spatial resolution of the climate datasets used. Please provide this information. In addition, it only becomes clear in Fig. 3 that the simulations extend until the year 2300— this should be stated explicitly in the Methods section.

Authors: We added mention of spatial resolution (which is 0.5° for W5E5 and varies between 1.12° and 2.50° for GCMs) of climate data in section 2.4, as well as the simulations' extent at L22.

**Section 3 (Results):**

• Figure 2 caption: Suggest rephrasing as: "Timing and runoff at peak water for varying ice volume fractions (a–b) and temperature biases (c–d)." Also specify directly in the axis labels that the multiplying factor applies to initial ice volume.

Authors: Agreed, we changed the caption to the suggested one and modified the axis labels of Fig. 2a and 2b to *'Initial ice volume multiplying factor''*.

• Figure 3 layout: Consider stacking the panels vertically. The phrase "the glaciers of the set" is unnecessary—if all 160 glaciers in the region are modeled, simply state that.

Authors: Agreed, Figure 3 was changed with vertically stacked panels. We replaced "the glaciers of the set" by "all 160 glaciers" in the caption.

• Figure 3 variability: The origin of variability in the light lines is unclear. If this reflects glacier altitude or location differences, please clarify; if not relevant, consider removing these lines.

Authors: Figure 3.d was changed and now represents the evolution of temperature and precipitation in the region under different scenarios (multi-GCM mean is shown in bold, shading is the mean ±1 standard deviation of the GCM ensemble).

• Sections 3.1–3.2: Both begin with descriptive sentences that duplicate figure captions. These could be removed to improve conciseness.

Authors: We removed descriptive sentences from beginnings of sections 3.1 and 3.2

- Wording corrections:
- "It appears clearly..." → "It is clear that" / "It clearly appears that."

Authors: Done.

"Indeed, increasing the total ice volume will not significantly advance" → remove "Indeed."

Authors: Done.

"We choosed to adjust" → "We chose to adjust."

Authors: Done.

• Literature placement: The statement "It is worth mentioning that previous work (Gao et al., 2010)" would fit better in the Discussion.

Authors: Changed this literature placement and added to the discussion at L273.

• Units: Use mm/yr or mm/day for precipitation, and retain m3/s for runoff, to align with hydrological conventions (Fig. 3 and text).

Authors: Done, units were changed in Figures 2 & 3, and in text as well.

• Supplementary material: Fig. S1 is only mentioned at the end of the Results without discussion; either integrate it into the narrative or remove it.

Authors: Supplementary material was added to the narrative in the revised version at L239-242:

"(...) as indicates the evolution of annual runoff decomposed by its four different contributions (Fig. S1). The latter highlights that under SSP1-2.6 and after 2150, runoff is mostly sustained by snow melt of ice-free areas with FARI19, whereas ice melt on glacier remains the largest component with MIL22. Under SSP5-8.5 annual runoff is largely due to precipitation on ice-free areas with both datasets".

**Section 4 (Discussion):**

Having a few subheadings would help to follow the discussion more easily

Authors: Agreed, we added four subheadings to the discussion: "Peak water dynamics and sensitivity", "Peak water dynamics and sensitivity", "Uncertainties and limitations in ice thickness data" and "Model limitations and implication for large-scale simulations".

• "Future work could explore more quantitatively the sensitivity" - should be "explore the sensitivity more quantitatively"

Authors: Done.

• The contrast between SSP1-2.6 and SSP5-8.5 is well described but somewhat repetitive of the Results section.

Authors: We reformulated the discussion section to avoid too much repetition with the results section at L245-272.

• The discussion of satellite missions and data sharing is interesting but feels speculative and less connected to the case study.

Authors: Agreed. We removed the sentences on future satellite missions and data sharing.

• While the study explores sensitivity to ice thickness and temperature, other sources of uncertainty (precipitation projections, model parameters, glacier dynamics, etc) receive limited attention and should be at least mentioned in the discussion.

Authors: Thank you for this comment. We added a sentence mentioning uncertainties related to glacier dynamics, precipitations and model parameters at L280-283.

**Technical and Formatting Issues:**

Minor errors ("choosed" → "chose"; "did not used" → "did not use").

Authors: Done

Several instances of comma splices

Authors: Done

Inconsistent use of past vs. present tense in methodology sections

Authors: Done

Elimination of wordy constructions and passive voice where possible

Authors: Done

---

## Author Comment (AC3)

**Reviewer 3**

In this study, Gimenes et al. use OGGM and two ice thickness products to assess the impact of geometric data and temperature bias on the magnitude and timing of peak runoff in the Western Kunlun Mountains. For future projections, the authors employ one GCM simulation extending to 2300 under SSP1-2.6 and SSP5-8.5 scenarios.

The topic is certainly interesting and relevant, but the study provides limited innovation in terms of methodology or analysis. The level of depth is rather modest, and several aspects of the methods and datasets require clarification. Even for a Brief Communication, I believe these points should be in depth addressed. While the results show that initial ice thickness strongly influences runoff, this outcome is somewhat expected. The "so-what" aspect of the study is currently missing and should be emphasised more clearly through deeper analysis and contextualization as well as richer figures. Given the potential of the topic, I recommend major revisions, with the hope that the comments below will help strengthen the manuscript.

Authors: We would like to thank Reviewer 3 their detailed comments that helped improving the paper clarity and quality.

**Major comments**

**• Introduction:**

The second part of the introduction does not really serve as an introduction to the study itself, but rather as a general overview of glacier modelling approaches. It would be more relevant to provide additional context on which variables most strongly influence glacier runoff and why ice thickness and temperature bias are key factors controlling peak water so that these are chosen. This could then naturally lead into the motivation and significance of the present study. Why is the Western Kunlun Mountains are chosen should also be addressed.

Authors: We provided more information on parameters controlling glacier runoff in the second paragraph of the introduction at L18-20. The choice of the region was addressed in the introduction of the revised version (see earlier comments to Reviewer #1 & #2).

**Methods**

- The authors mention that the ice thickness data from Millan et al. (2022) were corrected to match the consensus estimate (reported for 2003), but the procedure is not entirely clear. Was the ice thickness change from 2000–2019 (Hugonnet et al., 2021) applied across the full domain, or only partially (e.g. 2003–2017)? It would also be helpful to clarify what the reported average value of 4.0 km³ (line 90) represents.

Authors: The correction procedure was further explained at L91-97; ice thickness change was applied on the full time domain 2000-2019 and we added percentage of change in ice mass compared to original estimates (see response to comments of reviewer #2):

"we corrected MIL22 estimates to get corresponding thicknesses in the year 2000. As correction, we simply subtracted average glacier thickness changes of Hugonnet et al. (2021) between 2000 and 2019, which is obtained from DEM differencing, to MIL22 thicknesses. While solving temporal ambiguities of the thickness models is complicated, since DEM sources and in-situ data are not properly dated, this correction may be a step toward roughly matching the timing of the consensus estimate. Overall, differences after correction for glacier mass change average 6 km3 compared to the original MIL22 estimate, representing 1 % of the latter."

- The calibration section needs revision. It currently reads as a mix of "limitations" and "comparison with previous studies", which weakens the message and the perceived robustness of the study. Why not rerunning OGGM using the new configuration to ensure consistency with the latest developments?

Authors: We are now using the new configuration of OGGM for the revised version, so the calibration section was accordingly shortened.

- The discussion of the temperature bias (lines 180-181) is too brief, as it is a central element of the title. Is this bias the same as that derived during the calibration? Was the same procedure repeated but with a uniform bias applied over the full simulation period? Please clarify this.

Authors: We clarified the experiment at L172-175 as following:

"To this aim, we conceive an ensemble of projection runs (also using MIL22 dataset) adding a uniform temperature bias (ranging from -5 to 5°C) over the full simulation period, meaning that a this bias is added to the temperatures time series used by the model to calculate surface mass balance (additionally to any bias calculated for the mass balance calibration, see section 2.3)."

**· Results and discussion:**

- Figures 2 and 3 could be further optimised. Axis limits are inconsistent, which makes comparison difficult. Colour choices are sometimes awkward; using the IPCC standard colour palette would improve clarity. Labels are not always self-explanatory, making the caption crucial for interpretation, and overall label thickness should be increased for readability.

Authors: Figures 2 & 3 were further optimized accordingly to the comment, improving overall clarity. However, we cannot apply the exact IPCC colors for SSPs since one of the co-authors has color-blindness.

- The discussion lacks comparison with previous studies or observational runoff datasets that could provide at least partial validation. As a result, the broader context and implications of the results remain unclear.

Authors: As detailed above, the purpose of this study is not to obtain an exact value of runoff, but rather to understand the sensitivity of peak water to geometry and to potential temperature biases that may be present in climate models. This objective is now clearly stated in lines 32–38. Note, however, that we have nevertheless included a comparison with runoff estimates in the Tarim region (Gao et al., 2010) for the period 1961–2006 in lines 273–279.

**Minor comments and suggestions:**

**Abstract**

• Line 1: Sensitivity to what? Please specify (-> ice thickness and temperature?).

Authors: lce thickness and temperature are already mentioned in the following sentence and adding it again would be too repetitive.

• Line 4: Replace Predicted with Projected.

Authors: Done.

**Introduction**

Line 11: Add a reference for the Sea Level Rise component.

Authors: The reference is at the end of the sentence of L11, so there is no change in the revised version.

Line 12: Remove also.

Authors: Done.

**Data and Methods**

• Line 37: Replace with "more precisely, the Western Kunlun Mountains." (or West?)

Authors: Done.

• Line 38: Clarify why RGI v6.0 is used to define the study area (maybe better add this reference in the next sentence).

Authors: RGI v6.0 was used in Farinotti et al., 2019 and Millan et al., 2022 to invert ice thickness. It is therefore used again in this study. A clarification and reference was added at L43: "This study specifically targets a group of 160 glaciers with a total surface area of approximately 2900 km2, calculated with the RGI v6 (RGI Consortium, 2017) (Fig. 1)."

• Line 45: The phrase "these glaciers" currently implies 10% of 70%. Please reword for clarity.

Authors: Agreed, we clarified that by replacing "these glaciers" by "the remaining 20%" at L50.

• Line 50: A thinning rate of -9.6 m yr-1 seems unrealistically high? Please verify the source or units.

Authors: We double checked this value and realized we had read the wrong rate. Mean elevation change rate is now of  $-0.23 \pm 0.04$  m yr-1.

• Region 13 is generally thinning, but the Karakoram is an exception within RGI13, what does this imply for interpreting the Hugonnet (2021) regional value? Why not taking the Karakoram region from Hugonnet (2021) data separately?

Authors: We think that adding the Hugonnet et al. (2021) regional value provides context on the general thinning that is occurring at the scale of region 13, as a point of comparison with the Karakoram anomaly illustrated by values from Brun et al. (2017) previously mentioned in the text (L53-54). There is no mentioned value for only the Karakoram in the published paper of Hugonnet et al. (2021); the acceleration of thinning in this area is mainly illustrated in extended data Figure 7 of the same study.

• Figure 1: The inset zoom on the study area in HMA is too small. Please revise and consider adding the ice thickness dataset already in the panel (not only in the caption). The figure should be interpretable without relying on the caption. If ice thickness measurements exist in the study region, they could be included.

Authors: We have added the source of each dataset inside the Figure 1. The inset panel is already hiding some glaciers of the region, and is only meant to provide the rough location of the mountain range.

• Line 63: Add a reference for this statement. This could also be moved to the introduction to justify the regional focus.

Authors: We repeated the reference for this statement at L66. Concerning moving to the introduction, we have already accounted, in response to earlier comments which leading to a reshaping of the Introductory part.

• Line 67: Replace Predicted with Projected or Simulated.

Authors: Done, replaced predicted with simulated.

• Line 68: The consensus (or community) estimate was published in 2019 but represents 2003 (I thought actually more 2000), not 2019.

Authors: Agreed, see response to similar comment by Reviewer #2.

• Line 73: Clarify the phrase "the use of glacier by."

Authors: We clarified that with the phrasing "the use of flowline inversions "glacier by glacier"" at L77-78.

• Line 84: Confirm whether Farinotti et al. (2019) used measurements.

Authors: We added information about the use of measurements in Farinotti et al. (2019) from the GlaThiDa v2 dataset at L87-88: "Farinotti et al. (2019) used thickness measurements from the Glacier Thickness Database (GlaThiDa) v2 (WGMS, 2016), with less than 50 glaciers being covered within this RGI region".

• Lines 84-89: The correction method for the Millan dataset is unclear — was the Hugonnet thickness change simply added?

Authors: We clarified the correction, see response to major comment on this issue.

• The OGGM initialization description is too detailed for a Brief Communication. Focus on "We use OGGM, a model..." and emphasize what was new or modified for this study.

Authors: We shortened the section with the new model framework, see response to Reviewer #2's comment on this issue.

• Lines 108–112: Clarify which dataset was ultimately used and for what time period.

Authors: We clarified in section 2.4 'Climate forcing' that W5E5 data is used for dynamical spin-up or historical runs and that CMIP6 GCMs are used to simulate glaciers evolution from 2020 until 2300.

• Lines 113–127: This section reads more like discussion than methods. It could be shortened substantially. Based on line 121, it even seems that part of the research question has already been answered.

Authors: This section was removed as we are using a new model framework in the revised version, calibration section is now at L108-110.

• Lines 154–159: This belongs more logically in the description of the reanalysis dataset. Ensure consistent verb tense (past or present) throughout each section.

Authors: This section was also removed since we now use the dynamical spin-up procedure in the revised version.

• Consider first presenting the individual glacier runoff results before aggregating to the regional scale.

Authors: We added a description of an individual glacier runoff composition at the end of section 2.3' OGGM', L116-118:

"Finally, the model gives as simulation outputs glacier volume, length and area, as well as glacier runoff. Considering a fixed glacier area including glacierized and increasingly non-glacierized terrain, the annual total runoff computed in OGGM is derived as the sum of snow melt on now ice-free area, the ice and

seasonal snow melt on glacier, the liquid precipitation on glacier, and the liquid precipitation on now ice-free area (e.g. in Fig. S1)."

• There is some mixing of climate data, geometry, and model description; please reorder for clarity.

Authors: We did reorder the section order for clarity in the revised version: 2.3 OGGM, 2.4 Climate forcing, 2.5 Integration of ice thickness dataset and 2.6 Peak water calculation.

• Why was only one GCM used, given that six are available? The paper itself acknowledges that using a single GCM limits accuracy — please justify this choice.

Authors: We are now using a multi-GCMs ensemble, see response to comment of Reviewer #1 for more details.

**Results**

• Figure 2: It is unclear what the "multiplication factor" represents. Also, clarify the meaning of "temperature bias." Ensure consistent y-axis ranges across panels and apply IPCC colours for SSP1-2.6 and SSP5-8.5.

Authors: We changed "multiplication factor" to "initial ice volume multiplication factor". We think that the signification of temperature bias was further explained in section 2.4 (see response to previous comment). y-axis are now consistent across panels of figure 2 and we also changed the color palette.

• Figure 3: Move the figure closer to the relevant text. The colour scheme used to distinguish Farinotti et al. (2019) and Millan et al. (2022) is identical to that used for SSP scenarios in Figure 2 — please revise for clarity. Label font size should be increased.

Authors: Agreed. We moved the figure closer, label size was increased and color scheme changed. We did not use the exact IPCC colors for SSPs because it made it difficult to read for one of our color-blind co-authors.

• The content of panel C is not intuitive, why not show annual mean temperature and total precipitation instead?

Authors: The corresponding panel was changed in the revised version and is now temperature and precipitation from the different used GCMs.

• Lines 241-246: Move to the discussion section.s

Authors: Done.

**Discussion**

Consider subdividing this section into subsections for clarity.

Authors: Done, discussion was divided into four subheadings.

• Is Millan et al. (2022) systematically thicker across the entire glacier, or mainly in the accumulation area? If the latter, would that difference significantly affect runoff?

Authors: MIL22 is thicker than FARI19 across the entire glacier, both at low and high elevations (see Figure 1c). We added a sentence to describe this at L97-99.

• Line 280: Please clarify what "sufficient" refers to.

Authors: "sufficient" was replaced by "warm enough" in this paragraph now at L257.

• Lines 297-304: The outlook is vague. The discussion should place the results in a broader context and compare them with previous studies. Many results are rather intuitive (e.g. thicker ice leads to delayed and higher peak water). It would be more appropriate to revisit the research questions rather than expand on new remote-sensing products, which remain uncertain if and when they will come.

Authors: In response to an earlier comment, the discussion on data sharing and satellite mission is now removed. We further disagree with this comment since the discussion already placed the study in a broader context. We start by (1) providing interpretation to the sensitivity analysis results, (2) compare the calculated runoff with other studies on longer time period, (3) discussing limitation of the methodology used to assimilate ice thickness in global models, (4) discuss the broader context of ice thickness uncertainties and global model geometry limitations. No changes.

• Why is there no conclusion section? Please include one to summarise the key findings and implications, even in a couple of sentences.

Authors: We added a conclusion in the revised version:

"This study highlights the strong sensitivity of peak water timing and magnitude to uncertainties in initial glacier thickness and temperature biases in climate models. In regions where ice thicknesses are highly uncertain, such as the Western Kunlun mountains, peak water can be delayed by a decade, while its magnitude can change by up to 27% depending on the data source used under SSP-5.8.5. With the same scenario, peak water date can be brought forward by roughly a decade for each degree of temperature bias in the climate forcing data used. Finally, our results emphasize that accurate estimates of glacier geometry are crucial for robust projections of future water availability."